# Damage-Associated Molecular Patterns (DAMPs) in Retinal Disorders

**DOI:** 10.3390/ijms23052591

**Published:** 2022-02-26

**Authors:** Binapani Mahaling, Shermaine W. Y. Low, Molly Beck, Devesh Kumar, Simrah Ahmed, Thomas B. Connor, Baseer Ahmad, Shyam S. Chaurasia

**Affiliations:** 1Ocular Immunology and Angiogenesis Lab, Department of Ophthalmology and Visual Sciences, Froedtert and MCW Eye Institute, Medical College of Wisconsin, Milwaukee, WI 53226, USA; bmahaling@mcw.edu (B.M.); wlow@mcw.edu (S.W.Y.L.); mobeck@mcw.edu (M.B.); dkumar@mcw.edu (D.K.); siahmed@mcw.edu (S.A.); tconnor@mcw.edu (T.B.C.); bahmad@mcw.edu (B.A.); 2Vitreoretinal Surgery, Froedtert and MCW Eye Institute, Medical College of Wisconsin, Milwaukee, WI 53226, USA; 3Department of Cell Biology, Neurobiology and Anatomy, Medical College of Wisconsin, Milwaukee, WI 53226, USA

**Keywords:** DAMPs, endophthalmitis, uveitis, glaucoma, ocular cancer, ischemic retinopathies, diabetic retinopathy, age-related macular degeneration, proliferative vitreoretinopathy, inherited retinal disorders

## Abstract

Damage-associated molecular patterns (DAMPs) are endogenous danger molecules released from the extracellular and intracellular space of damaged tissue or dead cells. Recent evidence indicates that DAMPs are associated with the sterile inflammation caused by aging, increased ocular pressure, high glucose, oxidative stress, ischemia, mechanical trauma, stress, or environmental conditions, in retinal diseases. DAMPs activate the innate immune system, suggesting their role to be protective, but may promote pathological inflammation and angiogenesis in response to the chronic insult or injury. DAMPs are recognized by specialized innate immune receptors, such as receptors for advanced glycation end products (RAGE), toll-like receptors (TLRs) and the NOD-like receptor family (NLRs), and purine receptor 7 (P2X7), in systemic diseases. However, studies describing the role of DAMPs in retinal disorders are meager. Here, we extensively reviewed the role of DAMPs in retinal disorders, including endophthalmitis, uveitis, glaucoma, ocular cancer, ischemic retinopathies, diabetic retinopathy, age-related macular degeneration, rhegmatogenous retinal detachment, proliferative vitreoretinopathy, and inherited retinal disorders. Finally, we discussed DAMPs as biomarkers, therapeutic targets, and therapeutic agents for retinal disorders.

## 1. Introduction

Damage-associated molecular patterns (DAMPs) are endogenous danger molecules released from the extracellular and intracellular space of the damaged tissue or dead cells [1]. DAMPs are (i) rapidly released following necrosis; (ii) produced by the activated immune cells via specialized secretion systems or by the endoplasmic reticulum (ER)—Golgi apparatus secretion pathway; (iii) known to activate the innate immune system by interacting with pattern-recognition receptors (PRRs), and thereby directly or indirectly promote adaptive immunity responses; (iv) inclined to contribute to the host’s defense and pathological inflammatory responses in non-infectious diseases; and (v) responsible for restoring homeostasis by promoting the reconstruction of the tissue [1,2]. Accumulating evidence indicates that DAMPs are associated with the sterile inflammation caused by aging, increased ocular pressure, hyperglycemia, oxidative stress, ischemia, mechanical trauma, stress, environmental condition, and genetic defects during retinal development [3,4,5,6]. Recent studies suggested that DAMPs that include extracellular matrix pro (ECM)-proteins such as decorin, biglycan, versican, aggrecan, phosphacan, low-molecular-weight (LMW) hyaluronan, heparan sulfate (HS), fibronectin, laminin, tenascin-C, and tenascin-R; cytosolic proteins such as leukemia inhibitory factor (LIF), S100 proteins, uric acid, heat-shock proteins (HSP), adenosine triphosphate (ATP), cyclophilin A, F-actin; those of nuclear origins such as histones, high-mobility group box 1 (HMGB1), high-mobility group nucleosome binding domain 1 (HMGN1), interleukin (IL)-1α, IL-33, surface-interacting 3A (Sin3A)-associated protein 130 (Sap130), deoxyribonucleic acid (DNA), and ribonucleic acid (RNA); those of mitochondrial origins such as mtDNA, transcription factor A mitochondrial (TFAM), formylated peptides, mitochondrial reactive-oxygen species (mtROS); those of endoplasmic reticulum (ER) origins such as calreticulin, defensins, cathelicidins (LL37), endothelin-1 (ET-1) and granulysin; those of plasma membrane origins such as syndecans, glypicans, perlecan; and plasma proteins such as fibrinogen, Gc-globulin, and serum amyloid A (SAA), are increased; this suggests a protective or pathogenic role in different retinal disorders [1,7,8,9]. DAMPs function through multiple specialized innate immune receptors, such as receptors for advanced glycation end products (RAGE), toll-like receptors (TLRs) and the NOD-like receptor (NLRs) family, purine receptor 7 (P2X7), NLR pyrin domain 3 (NLRP3), in retinal disorders [1,10,11,12].

The eye is an immune privilege tissue and limits its local immune and inflammatory responses to preserve vision. Though the mechanism of immune privilege is not entirely understood, the tear-fluid barrier, epithelial barrier, blood–ocular barrier, and the inner and outer blood–retinal barriers play essential roles in the immune responses of the eye [13,14,15]. The retinal cells that play a regulatory role in the posterior segment of the eye are retinal pigment epithelial (RPE) cells which express Fas ligand and programmed death-ligand 1 (PDL1), and microglia/macrophages expressing regulatory elements such as CD200/C200R, PDL1, and T_reg_ cells. The anterior and posterior segment of the eye contains immunosuppressive fluid containing neuropeptides such as transforming growth factor-β (TGF-β), vasoactive intestinal peptide (VIP), somatostatin, calcitonin, gene-related peptide, alpha-melanocyte-stimulating hormone, neuropeptide Y, and pigment epithelial-derived factor (PEDF) [13,14]. Any perturbations in the retinal microenvironment are recognized by astrocytes and microglia present at the forefront of the defense system. Perturbations can arise from two major sources: (i) microbial pathogens and (ii) age- or disease-related injury. Astrocytes and microglial cells possess signaling mechanisms for host defense that are activated by recognizing structural characteristics found in pathogens, known as pathogen-associated molecular patterns (PAMPs) and DAMPs [4].

The innate immune system provides the first line of defense against the DAMPs. In the early stages of retinal disorders, microglia and the complement system activate at low levels. This low level of inflammation is essential to maintain homeostasis and restore functionality in retinal homeostasis. However, prolonged insult and stimulation by DAMPs in chronic retinal disorders such as glaucoma, age-related macular degeneration (AMD), diabetic retinopathy (DR), ischemic retinopathies, and uveitis lead to maladaptation of the innate immune system and dysregulated inflammation. As a result, increased pro-inflammatory cytokines such as tumor necrosis factor (TNF)-α, IL-1β, IL-6, and IL-8 contribute to further progression of the disease. Finally, immune privilege is compromised in retinal disorders, resulting in a vicious cycle of inflammation, leukocyte infiltration, and retinal neurodegeneration.

## 2. DAMPs in Retinal Disorders

### 2.1. DAMPs in Endophthalmitis

Endophthalmitis is a devastating and potentially blinding disorder caused by an infection from exogenous or endogenous microorganisms, typically in the vitreous cavity of the eye [16]. The inflammatory component in endophthalmitis is strongly associated with the recognition of microorganism PAMPs and damaged or dying cell DAMPs by TLRs located on the cell membrane and within endosomes [17,18]. *Staphylococcus aureus* (*S. aureus*) infection significantly enhances the expression of DAMPs such as S100A7/S100A9 in the retina. DAMPs released by the neutrophils provide a host-defense response but activate an inflammatory feedback loop when released to the extracellular surface [18]. In endophthalmitis patients, increases in vitreous HMGB1 directly correlates with the duration of infection and reduction in visual acuity [19,20]. HMGB1 function can vary based on its location. In the nucleus, HMGB1 binds to DNA and controls transcriptional regulation. On the other hand, HMGB1 can be passively released into the extracellular space by necrotic cells and activated macrophages, initiating a pro-inflammatory cytokine-like response [20]. The various DAMPs described in endophthalmitis are mentioned in Table 1.

In *S. aureus*-induced endophthalmitis, there is a significant increase in small HSP and αβ-crystallin in the retina. This prevents apoptosis of retinal cells and tissue destruction during immune clearance of the bacteria [21]. Additionally, a significant increase in LIF has been reported in the retina after *Bacillus cereus*-induced endophthalmitis. Although the precise role of this increase in LIF is not known, it was speculated to have a protective role in the retina [22]. These endophthalmitis patients also showed a significantly higher level of IL-1α concentration in the vitreous compared to the control subjects. Given that the IL-1 family plays a vital role in pathogen recognition, it stands to reason that the significant increase might have a protective role [23]. 

Defensins are cationic antimicrobial peptides that display antibacterial activity against Gram-positive and Gram-negative bacteria, fungi, and viruses [24]. In the human eye, two types of defensins are secreted: α-defensins released by peripheral mononuclear leukocytes (PMNs) within the ocular mucosa and tears, and β-defensin-1 secreted by the cornea and conjunctiva. Both are found in the aqueous and vitreous humor in the eye. In contrast, β-defensin-2 is not constitutively present, but is released in states of inflammation or infection. β-defensin-2 is secreted by RPE, ciliary body epithelium (CBE), and Müller glial cells. Interestingly, it has a regulatory element, nuclear factor kappa B (NFκB), and may act through the NFκB signaling pathway [24,25]. Post-microbial infection, the Müller glial cells secrete cathelicidin LL37, an antimicrobial peptide that plays an essential role in the innate immune response to endophthalmitis. Cathelicidin LL37 inhibits biofilm formation and is involved in chemotaxis, angiogenesis, and wound healing [25]. Cathelicidin LL37 greatly enhances cells response to self-nucleic acids released from damaged and dying cells. Cathelicidin LL37 peptide disrupts immune tolerance towards nucleic acid, permitting recognition by intracellular recognition systems such as TLR3, TLR7, TLR8, TLR9, mitochondrial antiviral-signaling protein (MAVS), and stimulator of interferon genes (STING) [26]. Additionally, SAA levels are increased significantly in infectious endophthalmitis patients, suggesting SAA as a potential biomarker for endophthalmitis [27].

### 2.2. DAMPS in Uveitis

Uveitis is an acute, recurrent, and chronic inflammation of the uvea caused by the breakdown of the immunosuppressive intraocular microenvironment [28]. Uveitis is characterized by compromised blood–ocular barriers, cellular infiltration, and tissue damage [29]. As a result, inappropriate intraocular inflammation can be detrimental to the eye and its visual function. DAMPs play a significant role in non-infectious uveitis by activating PRRs and TLRs, thus initiating an acute inflammatory response [28]. The different DAMP molecules increased in uveitis are S100 proteins, HMGB1, HSP70, SAA, fibronectin, and fibrinogen, as mentioned in Table 2. 

S100 proteins play an essential role in uveitis inflammation. Increased levels of S100A8/A9 and S100A12 were reported in the serum and aqueous humor of patients with autoimmune uveitis. S100A12 was found to be increased in the tear fluid of uveitis patients [30] and is actively secreted by the phagocytic cells upon cell activation. Once secreted, S100A8/A9 and S100A12 act as pro-inflammatory ligands and bind to TLR4 or RAGE, triggering inflammatory pathways [36]. Retinal cells also release HMGB1 in uveitis [31]. Usually, HMGB1 is secreted by macrophages during cellular stress or necrosis and mediates its actions as a DAMP through RAGE, TLR2, and TLR4 receptor signaling. HMGB1 recruits inflammatory cells and amplifies the local inflammatory response by inducing pro-inflammatory cytokines such as TNF-α, IL-1, and IL6 [37]. 

Serum uric acid levels are increased in many inflammatory conditions in the eye, including uveitis. Uric acid triggers endothelial dysfunction, oxidative stress, inflammation, and microvascular disease. However, the study did not find any significant increase in serum uric acid in uveitis patients [6]. The serum concentration of HSP70 has been reported to be enhanced in patients with concurrent Behcet’s disease and uveitis relative to those without uveitis [32]. When released to the extracellular space from the necrotic cells or cells under stress, HSPs act as DAMPs on multiple receptors such as TLR2 and TLR4. Additionally, they activate the NFκB signaling pathway in macrophages and dendritic cells to stimulate the production of cytokines and chemokines, thereby mediating the uptake and presentation of peptides via the major histocompatibility complex (MHC) to facilitate cell migration [28,37,38]. Furthermore, HSP-derived peptides 336 – 351 induce clinical and histological characteristics of uveitis in 80% of rats [39]. HSP90 inhibitors showed promising results in ameliorating experimental uveitis through the inhibition of NFκB, hypoxia induced factor (HIF)-1α, p38, and phosphatidylinositol 3-Kinase (PI3K) activity, and a reduction in vascular endothelial growth factor (VEGF), TNF-α, and IL-1β levels [38,39,40]. 

SAA is an acute-phase protein found in increased levels in the systemic circulation during chronic inflammatory disorders. Patients with uveitis or juvenile idiopathic arthritis with chronic anterior uveitis had higher SAA levels than their respective controls in the aqueous humor [33,41]. SAA acts on TLRs, NFκB, and P2X7-dependent NLRP3 inflammasome in macrophage and antigen-presenting cells (APCs), thus playing an essential role in inflammatory cytokine production, neutrophil transmigration, monocyte migration, and peripheral blood mononuclear cell (PBMC) adhesion and differentiation [42,43,44]. Though IL-33 acts as a DAMP, it has an anti-inflammatory effect for its role in activating M2 macrophage polarization and attenuating the development of experimental autoimmune uveitis [45]. Additionally, the intraocular cellular fibronectin levels were significantly higher in patients with active uveitis [34]. In another study, the concentrations of fibronectin, fibrinogen, and immunoglobulins were significantly higher in the iris of the uveitis subjects compared to the controls. The irises of patients with uveitis also showed higher T-lymphocytic infiltration. These findings suggest that the presence of fibronectin, fibrinogen, and immunoglobulins significantly contribute to T-lymphocyte infiltration and inflammation in uveitis [35].

### 2.3. DAMPs in Glaucoma

Glaucoma is a neurodegenerative disorder that causes damage to the optic nerve axons, resulting in the loss of retinal ganglion cells (RGC). The major risk factors for glaucoma include aging, family history and genetics, and intraocular pressure (IOP) elevation. Strong evidence suggests that an early insult to RGC axons at the optic nerve head may involve astrocytes, microglia, and other blood-derived immune cells [4]. DAMPs are also involved in glaucoma. The DAMPs produced and identified in glaucoma are mentioned in Table 3.

S100B was co-localized with astrocytes and Müller glia in the autoimmune glaucoma rat model [46]. S100B activates pro-inflammatory cytokines, such as IL-1β and TNF-α, and stress-induced enzymes, such as nitric oxide synthetase, potentially resulting in ganglion cell death [66]. Similarly, immunization with S100B leads to ganglion cell death, indicating its involvement in neuroinflammation [66]. In acute ocular hypertension, LIF and LIFR were significantly increased in the retina. The study suggested that LIF may be critical for the process of degeneration/protection following retinal ischemia via activation of the Janus kinase (JAK)/STAT and Akt signaling pathways [47]. In fact, a neuroprotective role is postulated based on observations following intravitreal injection of LIF [67]. Serum uric acid levels were also increased in primary open-angle glaucoma patients compared to the control group [48]. The increase in uric acid concentration was also reported in aqueous humor of subjects with glaucoma [68]. On the contrary, lower serum uric acid concentration was observed in primary angle-closure glaucoma in another study. Further, its negative association with disease severity suggests uric acid as an important candidate in response to glaucoma-associated oxidative stress [69].

In previous studies, HSPs were increased in response to elevated IOP, as also seen in human glaucomatous retinas [4,49]. Immunization with HSP27 and HSP60 led to pressure-independent RGC degeneration and axon loss, mimicking glaucoma-like damage [70]. These findings indicate that HSPs in glaucoma may be directly involved in disease onset and glaucoma progression. Notably, there was a significant increase in ATP in the aqueous and vitreous humor of patients with primary open-angle glaucoma. The activation of P2X7 by ATP elevates intracellular calcium, resulting in rat RGC death [50]. In addition, significantly high levels of amyloid beta (Aβ) have been reported in the optic nerve head and aqueous humor of glaucoma patients [51]. Aβ co-localizes with apoptotic RGC in the experimental glaucoma rat model and induces significant RGC apoptosis in vivo in a dose- and time-dependent manner [71]. Additionally, intraocular injection of Aβ1–40 appeared to have a time- and dose-dependent effect on neurodegeneration with increased axonal swelling and RGC cell death, leading to ganglion cell layer (GCL) thinning and optic nerve injury [72]. The activation of Aβ may lead to activation of neuroinflammatory pathways, and hence, glaucoma progression with or without IOP-elevation-related triggers [73]. There was also a significant increase in autoantibodies against Histone H4 in the serum of glaucoma patients [52]. However, the precise role of Histone H4 in glaucoma is not known. 

HMGB1 concentrations were significantly higher in the aqueous humor of primary open-angle glaucoma patients, whereas in rodents, HMGB1 was linked to glaucoma induced by elevated IOP. HMGB1 significantly upregulates canonical NLRP3 inflammasome via caspase-1 and non-canonical caspase-8-driven inflammasome, which results in IL-1β release, thereby causing ganglion cell death [53,74]. Additionally, IL-1α concentrations were noted to be significantly increased in the aqueous humor of primary open-angle glaucoma with and without diabetes [54]. Furthermore, there was a significant increase in nuclear and mitochondrial DNA damage during ganglion cell death [55,75]. However, the exact role of extracellular DNA released from dead cells in glaucoma has not been described. 

The progressive retinal atrophy (PRA)1 family protein 3, calnexin, calreticulin, clusterin, 78 kDa glucose-regulated protein, heterogeneous nuclear ribonucleoprotein R, malectin, peptidyl-prolyl cis–trans isomerase B, protein disulfide isomerase, reticulocalbin 3, and heterogeneous nuclear ribonucleoprotein Q, were reported to be significantly high in a non-human primate model of early experimental glaucoma [56]. However, the role of ER stress in glaucoma has not been studied yet. Optic nerve astrocytes proliferate after treatment with ET-1 (also known as EDN1), and reactive astrocytes increase endothelin receptor B (ETB) expression in both human and experimental neuronal injury models. Increased expression of ET-1 causes vasoconstriction, which prevents the optic nerve vasculature from responding to the need for increased blood flow. Hence, ET-1 could be central to autoregulatory disturbances in glaucoma [57]. Additionally, ET-1 causes neuronal cell death in glaucoma by activating pro-apoptotic transcription factor JUN (the canonical target of JNK signaling) [76]. 

The small, leucine-rich proteoglycan (SLRP) family of DAMP proteins has been suggested to play a critical role in glaucoma. Decorin concentrations decreased significantly in the aqueous humor of primary open-angle glaucoma patients [58]. Intracameral injection of recombinant human (rh) decorin decreased TGF-β -induced fibrosis, lowered IOP, and prevented ganglion cell loss [77]. Another SLRP, biglycan, was significantly increased in the optic nerve head of non-human primates in early experimental glaucoma, indicating its role in disease progression [59]. Versican, a large proteoglycan, may organize glycosaminoglycans (GAGs) and other ECM components to facilitate and control open flow channels in the trabecular meshwork, which appear to be a central component of the outflow resistance [60]. Interestingly, a significant decrease in aggrecan was found in the optic nerve head of glaucomatous eyes compared to control eyes of non-human primates [61]. However, such findings were absent in rodent models [62]. A significant increase in phosphacan levels was also observed in an autoimmune glaucoma rat model, with studies indicating its role in disease progression [62,78]. There was a significant increase in chondroitin sulfate and HS in serum and optic nerve heads of glaucoma patients [63]. In a mouse glaucoma model, increased fibronectin, laminin, and tenascin-C levels were also found in the glaucomatous heterozygous retina and optic nerve compared to the wild-type group [62]. Fibronectin was explicitly found at higher levels in the trabecular meshwork of glaucomatous compared to non-glaucomatous eyes. This is significant, as elevated IOP results from increased ECM rigidity regulated by collagen IV and fibrillin deposition [64]. Tenascin-C is up-regulated in glaucomatous eyes, especially in astrocytes. As an endogenous activator of the TLR4, tenascin-C’s inflammatory role is being studied in glaucoma research [4,62]. SAA is also associated with glaucoma-related increased IOP and inflammation [65]. 

### 2.4. DAMPs in Ocular Cancer

Ocular cancers include retinoblastoma, uveal melanoma, and conjunctival melanoma [79,80]. Retinoblastoma is caused by sporadic somatic mutations in the RB1 gene, but about one-third of cases arise in infants with germline mutations [79]. Uveal melanoma is the second most common type of melanoma and arises from the melanocytes in the uveal tract. Conjunctival melanomas arise from melanocytes located in the basal layer of the epithelium in the conjunctival membrane [80]. The dysregulation of S100 proteins plays a vital role in growth, metastasis, angiogenesis, and immune evasion in cancer. The extracellular S100 proteins exert regulatory activities on microglia, neutrophils, lymphocytes, endothelial cells, neurons, and astrocytes [81]. Thus, they participate in innate and adaptive immune responses, cell migration, chemotaxis, and leukocyte and tumor cell invasion [82]. Retinoblastoma causes a significant increase in S100 protein in astrocytes, ganglion cells, and Müller glial cells [83]. More interestingly, the S100-positive cells have both neuronal and glial properties [84]. There is also a significant increase in S100 proteins in uveal melanoma [85]. A previous study compared S100A1 in paraffin-embedded sections of conjunctival naevi, conjunctival melanomas, and uveal melanomas. It was found that S100A1 was more frequently expressed in conjunctival and uveal melanoma than in conjunctival naevi [86]. S100B serum concentration was also significantly higher in uveal melanoma patients with metastases compared to uveal melanoma patients without, and may potentially be a future biomarker for metastatic uveal melanoma [87]. The distribution of various DAMPs found in ocular cancer have been summarized in Table 4.

Uric acid was elevated in the aqueous humor of eyes with melanoma, and in both the aqueous humor and tears of eyes with retinoblastoma [88]. The overexpression of HSPs provides a selective advantage to malignant cells by inhibiting apoptosis, promoting tumor metastasis, and regulating immune responses [94]. In control subjects, the human adult retina did not show HSP70/HSP90 immunoreactivity, whereas higher-to-moderate expressions of these proteins were observed in subjects with retinoblastoma tumors [89]. There was no significant difference in HSP27, HSP70, and HSP90 in uveal melanoma [95]. However, another study showed a higher degree of HSP90-positive staining in uveal melanoma cases, with 68% of cases staining positive and an average of 50% of tumor cells stained. The expression level was directly correlated with tumor diameter [96]. Additionally, extracellular vesicles derived from uveal metastatic melanoma have higher HSP70 and HSP90 than normal choroidal melanocytes, and more interestingly, these extracellular vesicles play an essential role in progression and metastasis [90]. 

Intracellular and extracellular HMGB1 has been implicated in tumor formation, progression, and metastasis. There is a significant increase in HMGB1 expression in retinoblastoma (RB) cells. HMGB1 levels have also been found to be significantly higher in human patient samples and associated with tumor differentiation and optic nerve invasion [97,98]. In the uvea, there is upregulation of HMGB1 with a binding affinity for the retinoblastoma tumor suppressor protein [91,99]. In patients with cancer, the circulating cell-free (cfc) DNA has the same genetic and epigenetic alterations compared to the related primary tumor. The majority of cfcDNA is derived from tissue tumor cells rather than from circulating tumor cells [92]. The aqueous humor of retinoblastoma patients contains tumor-derived cfcDNA, which can be used to diagnose the disorder [100]. The plasma cfcDNA can also detect somatic RB1 mutations in patients with unilateral retinoblastoma [101]. The blood plasma and aqueous humor of uveal melanoma patients also contain tumor-derived cfDNA which can be used for diagnosis [92,102]. The versican has been implicated in tumor progression, with abnormal mRNA expression observed in uveal melanoma. However, versican protein levels have not been reported [93]. 

### 2.5. DAMPs in Ischemic Retinopathies

Retinal ischemia occurs due to inadequate blood supply to the retina, required for oxygen diffusion and high metabolic activity. Circulatory failure can result from choroidal or retinal vessel obstruction. This lack of blood supply alters metabolic functions in the highly demanding retina and can ultimately result in irreversible neuronal cell death, vision loss, and blindness. Ischemic retinopathy causes include central retinal artery occlusion (CRAO), branch retinal artery occlusion (BRAO), central retinal vein occlusion (CRVO), branch retinal vein occlusion (BRVO), and DR. The location and level of obstruction to the blood supply determines the severity of ischemia, the area of retina affected, and its deleterious effects on the retina. DAMPs involved in ischemic retinopathies include S100 proteins, uric acid, HSPs, αβ-Crystallin, cyclophilin A, LIF, HMGB1, IL-1α, ECM proteins, and TFAM, which are summarized in Table 5.

A significant increase in the S100 protein in ganglion cells was reported in border zones damaged by retinal vein occlusion (RVO). However, this immunoreactivity was absent inside areas of completely non-perfused capillaries, indicating inflammatory recruitment of S100 proteins in RVO [103]. In addition, the expression of S100A4 was also found to be positively correlated with the progression of retinal neovascularization observed in oxygen-induced retinopathy (OIR) models [117]. Silencing S100A4 reduces brain-derived neurotrophic factor (BDNF) activation and VEGF expression, suggesting its role in regulating retinal neovascularization [117]. In addition, suppression of S100A4 can also reduce the expression of cAMP response element-binding protein (CREB) and B-cell lymphoma-2 (Bcl-2), and increase the expression of caspase-3, to promote apoptosis and prevent abnormal neovascularization [118]. Interestingly, overexpression of S100A4 provides neuroprotection in ischemic mice by activating the Akt pathway, thus suppressing apoptosis in RGCs [104]. Damage signals from S100A4 may influence diverse signaling pathways in different retinal cell types to elicit unique responses for protection against ischemia. The animal models that have been subjected to ischemia exhibit increased uric acid concentrations in the retina. Uric acid expression is transiently decreased following reperfusion, and subsequently increased in the later stages after 60 min [105,119]. Additionally, the oxidation of hypoxanthine and xanthine results in the production of uric acid during ischemic/reperfusion (I/R) injury [120]. 

Several HSPs, including HSP27, HSP70, and HSP72 play a role as DAMPs in the ischemic retina. HSP27 is a neuroprotective component that can be induced after acute pressure-induced ischemia [121]. During ischemic injury, its expression is upregulated in the neuronal and non-neuronal inner retinal layers [106,122]. Rats subjected to bilateral common carotid artery occlusion (BCCAO) displayed a significant increase in HSP27 and HSP70 immunoreactivity in the GCL after ischemic injury [107]. It was suggested that HSP27 might play a protective role in the retina. The delivery of HSP27 to RGCs via electroporation increased RGC survival rate after I/R injury [123]. In ARPE-19 cells induced with myeloperoxidase-mediated oxidative injury, HSP27 expression was increased, suggesting its role in the RPE injury response [108]. Similarly, HSP70 was also increased in rat retinas following I/R injury [124]. HSP-70 prevents apoptosis by upregulating Bcl-2 and interfering with apoptotic peptidase activating factor-1 (Apaf-1) to prevent apoptosome formation [125]. HSP72 expression has also been studied in ischemic retinopathy. The loss of retinal neurons in ischemic retinopathy is associated with glutamate-induced excitotoxicity. Intravitreal injection of a glutamate receptor agonist, N-methyl-D-aspartate (NMDA), can induce inner retina cell death. Post-NMDA injection, HSP72 expression was elevated in the retinal GCL [126]. The number of HSP72 stained RGCs was also significantly higher after acute pressure-induced retinal ischemia [121]. This study suggests that HSP72 can exhibit DAMP properties involved in the ischemic stress response. 

Cytoplasmic cyclophilin A plays a fundamental role in cell metabolism, and its expression levels can be altered in the presence of retinal lesions [109]. Rats exposed to more extended periods of ischemia exhibited a loss of circulating anti-cyclophilin A antibodies. These antibodies have been speculated to bind to damaged retinal tissues in response to ischemic injury. However, more analysis is required to determine the roles of cyclophilin A in ischemic retinopathy [127]. Aβ is another DAMP associated with neurodegenerative retinal disorders [128]. Production of Aβ is associated with neuronal apoptosis and cell loss. In primary retinal neuron cells treated with CoCl_2_ to induce hypoxia, Aβ expression was significantly increased, suggesting that Aβ may be altered during ischemic retinal damage [129]. LIF expression may also be altered in neuronal injuries and retinal disorders. LIF regulates gliosis and is a neuroprotective factor. Following retinal ischemia and retinal cell apoptosis induced by acute ocular hypertension, LIF and LIF receptor (LIF-R) expression were found to be increased, along with elevated levels of phosphorylated Akt [47]. LIF may modulate retinal injury and repair via the PI3K-Akt pathway. LIF can also inhibit retinal vascular development independent of VEGF, suggesting its role in vascular remodeling [130]. 

HMGB1 is a prototypic DAMP molecule localized to the GCL, inner nuclear layer (INL), and photoreceptor layer in the retina. It promotes inflammation, ganglion cell death, and photoreceptor degeneration in I/R-induced retinal damage [131]. Intravitreal injection of recombinant HMGB1 has been known to result in a loss of RGCs [110]. In vitro addition of HMGB1 to retinal glial cells also induced the production of pro-inflammatory factors [132]. However, the treatment of retinal ischemia with neutralizing anti-HMGB1 monoclonal antibodies has been controversial. One study found that intraperitoneal injection of a neutralizing anti-HMGB1 monoclonal antibody increased reactive oxygen species (ROS) production, resulting in retinal thinning and poor retinal function [133]. On the contrary, another reported that the neutralization of HMGB1 can prevent retinal thinning and loss of ganglion cells, and reduce the number of irregular retinal capillaries [134]. The differences in the neutralizing antibody concentrations could be a possible reason for these differing effects. IL-1α has also been shown to increase significantly in I/R-induced retinal injury [111]. Blood plasma cytokine analysis of rats with I/R injury presented elevated concentrations of IL-1α, TNF-α, and MCP-1 [112]. IL-1α gene expression has also been reported to rise rapidly, peaking at 3 to 12 h after rat retinal ischemia [135]. 

TFAM is a mitochondrial-DNA-binding protein crucial for mitochondrial gene expression and essential for oxidative phosphorylation-mediated ATP synthesis. TFAM protein expression significantly increases in the ischemic retina [113] and is localized to the outer plexiform layer (OPL), INL, inner plexiform layer (IPL), and GCL [114]. An increase in TFAM expression can prevent the alteration of mitochondrial DNA in the ischemic retina [113]. Preservation of TFAM may also promote an endogenous repair mechanism to protect RGCs against mitochondrial dysfunction during oxidative stress. In neonatal rat ischemic brain injury, TFAM protein expression was rapidly elevated and mitochondrial dysfunction and ROS generation were reduced [136]. TFAM expression during retinal ischemia may exhibit similar protective mechanisms. SAA, IL-6, and TGF-β are major proteins involved in the acute and chronic stages of inflammation. SAA is significantly higher in the aqueous humor of RVO patients with macular edema compared to controls [137].

The expression of extracellular glycoproteins decorin, fibronectin, laminin, tenascin-C, tenascin-R, and the chondroitin sulfate proteoglycans aggrecan, brevican, and phosphacan were studied in an ischemia-reperfusion injury model. Interestingly, decorin expression was reduced in the inner retinal layers in the early stages but increased substantially in the later stages of I/R, with strong immunoreactivity to damaged retinal layers. Fibronectin was significantly elevated in the retina following ischemia, while laminin, tenascin-C and aggrecan showed enhanced immunoreactivity in the optic nerve after ischemia, indicating their regulatory role during neurodegeneration [115]. Another proteoglycan, HS, can suppress aberrant neovascularization by inhibiting VEGF-A from binding to VEGF-R2 [116]. Fibronectin and tenascin-C expression were also increased, which localized to retinal blood vessels in the inner layers of the ischemic retina [3]. Since ECM proteins play an important role in vascular development and neovascularization, the upregulation of fibronectin in the ischemic retina could reflect its role in the remodeling of the retinal microvasculature. Elevation of tenascin-C concentrations can also contribute to retinal degeneration observed in ischemic retinopathy. In tenascin-C-deficient ischemic mice, ERG a- and b-wave amplitudes were higher than in wild-type ischemic mice [138]. Less rod photoreceptor degeneration was also observed in tenascin-C-deficient mice, suggesting that tenascin-C may be involved in ischemic retinal degeneration. Aggrecan and phosphacan are other extracellular DAMPs that have been studied in ischemic retinopathy. Protein expression of aggrecan and phophacan have been reported to be significantly reduced in the ischemic rat retina [115]. Downregulation of these DAMPs could be associated with retinal gliosis, reorganization, or the retinal degenerative process. 

### 2.6. DAMPs in Diabetic Retinopathy 

Diabetic retinopathy (DR) is a neurovascular retinal disorder in which inflammation and oxidative stress play a major role in disease progression [139]. DAMPs can sense high glucose as a stressor and directly corelate with the advancement of DR [5,140]. The different intracellular DAMP molecules increased in diabetic retinopathy are S100, HMGB1, uric acid, HSPs, ATP, cyclophilin A, Aβ, IL-1α, IL-33, nuclear DNA, mtDNA, mtROS, formyl peptide and lipid from mitochondrial membrane [5,140,141,142,143,144,145]. The list of DAMPs involved in the DR are mentioned in Table 6.

S100 proteins were found to increase in microglia and macrophage infiltration in the Akimba mouse model of proliferative DR [146]. Our study also reported an increase in plasma levels of S100A8 and S100A9 proteins in diabetic patients, which correlated with the severity of DR [5]. S100 proteins (S100A7, S100A12, S100A8/A9, and S100B) interact with RAGE and activate NFκB, inducing the production of pro-inflammatory cytokines and leading to the migration of neutrophils, monocytes, and macrophages [147]. In addition, HMGB1 is significantly increased in the vitreous humor of diabetic patients [148]. Similar to S100 proteins, HMGB1 can bind to TLR4 and RAGE, leading to increased inflammation via the NFκB pathway [140]. Uric acid, another DAMP, was also found to be elevated in the vitreous humor and serum of diabetic patients with macular edema [149].

**Table 6 ijms-23-02591-t006:** DAMPs in diabetic retinopathy.

Disease	DAMPs	Type	Origin	Localization
Diabetic Retinopathy 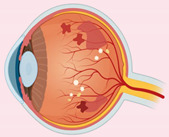	S100A8, S100A9 [5,146]	Ca^2+^ binding protein	Cytoplasmic	Macroglia/plasma
	HMGB1 [148]	Nuclear binding protein	Nuclear	Vitreous
	Uric acid [149]	Metabolic product	Cytoplasmic	Vitreous/perum
	HSP27, HSP60, HSP70 [150]	Molecular chaperones	Cytoplasmic	Retinal pndothelial cells
	ATP [144]	Nucleotide	Cytoplasmic	Microglia
	Cyclophilin A [151]	Ubiquitous protein	Cytoplasmic	Plasma
	Aβ [152]	Peptide	Cytoplasmic	RGC
	Calreticulin [153]	Multifunction soluble protein	ER	Plasma
	Cathelicidin [154]	Antimicrobial peptide	ER	Plasma
	α-defensin-1, -2, -3 [155]	Antimicrobial peptide	ER	Plasma
	Syndecan [156]	Proteoglycan	PM	Plasma
	Decorin [157,158]	Proteoglycan	ECM	Plasma/aqueous
	Versican [159]	Proteoglycan	ECM	Plasma
	LMW hyaluronan [160]	Glycosaminoglycan	ECM	Vitreous
	HS [161]	Glycosaminoglycan	ECM	Vitreous
	Fibronectin [34,162]	Glycoprotein	ECM	Plasma/vitreous/aqueous/retina
	Laminin [163]	Glycoprotein	ECM	Basement membrane/retina
	Fibrinogen [164]	Glycoprotein	ECM	Plasma
	Tenascin-C [165]	Glycoprotein	ECM	Vitreous

High glucose levels with elevated uric acid causes an increase in TGF-β, which plays an important role in retinal fibrosis in proliferative DR [166]. Uric acid increases the expression of Notch 1 receptors and ligands Dll1, Dll4, Jagged 1, and Jagged 2 in retinal endothelial cells, which promotes DR by increasing the activity of the Notch signaling pathway [141]. The overexpression and phosphorylation of HSPs affect vascular injury and neovascularization in DR [150]. Extracellular HSP70 binds with CD40 and TLR3, resulting in endothelial proliferation and migration, which plays an important role in retinal neovascularization [167]. Moreover, ATP released from damaged neurons and activated microglia acts as a pro-inflammatory molecule, initiating immunomodulatory, neurodegenerative, and hyperemic processes in the eye, which are mediated via activation of P2X7, P2Y1, and other ligand-gated P2X and G-protein-coupled receptor subtypes co-expressed in the retina [144]. Cyclophilin A is an important secreted oxidative-stress-induced factor, which is increased in the plasma levels of diabetic patients. It is secreted from endothelial cells and monocytes, and stimulates endothelial cell adhesion molecule expression to enhance the recruitment of circulating blood cells during the inflammatory response [151]. The secreted Cyclophilin A may also interact with the CD147 receptor of macrophage and induce the production of matrix metallopeptidase (MMP)-9 and pro-inflammatory cytokines to promote cell migration [168]. It plays an important role in blood–brain barrier repair, though the role of Cyclophilin A in DR is not yet known [151]. 

The diabetic retina indicates increased deposition of Aβ in the ganglion cells [152]. Aβ conciliates the RAGE-induced pro-inflammatory response via the TLR4 signaling pathway in the retinal ganglion cell line RGC-5 [169]. Moreover, hyperglycemia increases the production of Aβ and damages the endothelial tight junction by inhibition of zonula occludens-1 (ZO-1), claudin-5, occludin, and the junctional adhesion molecule (JAM)-C in endothelial cells [170]. In DR, there was no change in IL-1α expression, but there was upregulation of its receptor IL-1R in the diabetic retina. The nuclear translocation of IL-1α in the inner nuclear layer was higher in the diabetic retina compared to the non-diabetic control [171]. IL-1α is retained in the nucleus, tightly linked to chromatin, and released to the extracellular space after necrosis, but not by apoptosis. It interacts with IL-1R and activates MAPKs and NFκB, leading to the expression of pro-inflammatory cytokines, chemokines, and secondary mediators of the inflammatory response [172]. There was no observable significant difference in the levels of IL-33 in the serum, vitreous, or aqueous humor of proliferative DR patients. However, IL-33 is known to enhance M2 macrophage polarization in diabetic mice [173,174,175]. The nuclear and mtDNA released by the dead cells activate TLRs, NLRP3 and other cytosolic immune response platforms, which activates caspase-1 and the secretion of IL-1β [145]. Endosomal and lysosomal membrane-associated TLR9 can also bind to mtDNA to activate absent melanoma (AIM)2 inflammasome and caspase-1 [1,145]. In DR, damaged mitochondria release various DAMP molecules including mtROS, mtDNA, formyl peptides, and lipid components. The endoplasmic reticulum-based DAMPs, such as calreticulin, defensins and cathelicidin, are increased in plasma concentrations during diabetes [153,154,155], though only cathelicidin has been studied in DR. Under hyperglycemic conditions, calreticulin was observed to have higher expression in endothelial cells [176]. The plasma membrane-based DAMPs such as syndecans are significantly increased in the plasma of diabetic patients [156]. Syndecan-1 is known to inhibit leukostasis and angiogenesis by controlling leukocyte and endothelial cell interactions. Its increase in diabetes might play a protective role [177]. 

The ECM molecules, such as biglycan, decorin, versican, aggrecan, phosphacan, LMW hyaluronan, HS, fibronectin, fibrinogen, laminin, tenascin-C, and tenascin-R, are cleaved from the ECM and turned into a host-derived non-microbial DAMP [1,178]. Though the exact role of biglycan is not defined in DR, preliminary data suggest its angiogenic and inflammatory properties in DR [179,180]. Decorin concentrations have also been reported to be increased in the plasma of diabetic patients and the aqueous humor of DR patients [157,158]. Interestingly, decorin can be a multifunctional DAMP, acting on TLR2/TLR4 and TGF-β signaling pathways, deploying both pro- and anti-inflammatory effects [181]. In RPE, decorin prevents high glucose and hypoxia-induced epithelial barrier breakdown by suppressing p38 MAPK activation [182]. Plasma versican concentrations are also increased in diabetic patients [159], though its role in DR is not known. The increase in versican is associated with the invasion of leukocytes early in the inflammatory process. In addition, versican interacts with inflammatory cells either via hyaluronan or via CD44; P-selectin glycoprotein ligand-1 (PSGL-1); or TLRs present on the surface of immune and non-immune cells. These interactions are important for the activation of signaling pathways that promote NFκB, resulting in the synthesis and secretion of inflammatory cytokines such as TNF-α and IL-6 [183]. Aggrecan is produced by proteolytic degradation of the aggrecan core protein, and activates macrophages in a TLR2/myeloid-differentiation primary-response protein 88 (MyD88)-/NFκB-dependent manner, stimulating the expression of inducible nitric oxide synthases (iNOS), CCL2, IL-1α, and IL-6 [178]. The role of aggrecan in DR is not known, though its presence is increased in other ischemic conditions, as mentioned earlier. LMW hyaluronan is generated by the effect of free radicals, AGE products, and hyaluronidase enzyme activity, which leads to vitreous body liquefaction in DR [160,184]. These DAMPs stimulate endothelial cell proliferation, migration, and differentiation and may play a role in angiogenesis in proliferative vitreoretinopathy (PVR); they might also be the reason for proliferative retinopathy in diabetes [184]. Furthermore, LMW hyaluronan acts on CD44, TLR2, and TLR4 receptors and plays an important role in inflammatory pathways [185]. 

Interestingly, the soluble HS in the aqueous humor acts as a DAMP and shows an anti-angiogenic property by inhibiting the binding of VEGF to vascular endothelial cells. It inhibits pathological retinal angiogenesis in mice by inhibition of VEGF-VEGFR2 binding [116]. In younger individuals with diabetes, HS levels are low compared to older diabetic individuals, which provides a correlation between the higher susceptibility of younger subjects with diabetes mellitus and developing proliferative DR [161]. The intraocular and plasma concentration of cellular fibronectin increases in diabetes patients with macular edema [34,186]. In the early stages of DR, the deposition of fibronectin, collagen IV, and laminin, occurs in the endothelial basement membrane. The intravitreal injection of diabetic rats with antisense oligonucleotides to fibronectin, collagen IV, and laminin decreases hyperglycemia-induced vascular leakage [163]. The overexpression of fibronectin and laminin γ1 in the diabetic retina could also be correlated with enhanced TLR4 and P2X7 receptor levels in diabetic rats. This is in line with the activation of transcription factor NFκB, and histone H3 lysine 9 acetylation in diabetic retinas, which are implicated in proinflammatory gene induction [162]. Microglia can recognize the integrins α5β1 and α5β5 of fibronectin and become activated [187]. Fibrinogen results in microglia activation and CX3CR1-mediated inflammation in DR pathogenesis [188]. Plasma fibrinogen concentrations have also been reported to be directly corelated with the severity of DR [164]. Additionally, fibrinogen activates macrophages through TLR4 signaling and stimulates chemokine secretion [189]. The vitreous concentration of tenascin-C is highly correlated with proliferative DR [165]. Tenascin-C enhanced the sprouting, migratory, and survival effects of angiogenic growth factors, and had distinct proliferative, migratory, and protective capacities in vitro, and angiogenesis in vivo [190]. Tenascin-C activates TLR4 and induces soluble proinflammatory mediators, such as IL-6, IL-8, and TNF-α in microglia, macrophage, and dendritic cells [191,192]. 

### 2.7. DAMPs in Age-Related Macular Degeneration

Age-related macular degeneration (AMD) is a neurodegenerative disorder characterized by the accumulation of drusen (extracellular deposits) with the progressive destruction of photoreceptors and neural retina. AMD pathogenesis involves the metabolic abnormalities such as hypoxia, oxidative stress, and innate immunity responsible for the disease’s progression, ultimately leading to the loss of vision. AMD occurs predominantly in two forms, the atrophic or “dry” form and the neovascular or “wet” form. The various DAMPs involved in AMD pathogenesis are described in Table 7.

The activation of the innate immune system results in the release of DAMPs such as S100 proteins, uric acid, HSPs, ATP, Aβ, HMGB1, IL-1α, mtDNA, ET-1, and SLRPs. The vitreous samples of AMD patients showed higher extracellular ATP levels. In wet AMD with sub-retinal hemorrhage, the release of extracellular ATP induced severe photoreceptor cell death [193,194]. The extracellular ATP triggers an inflammatory cascade via TLRs and NLRs. Both TLRs and NLRs can trigger nuclear translocation of NFκB and subsequent transcription of IL-1β and IL-18 proinflammatory components and activation of the NLRP3 inflammasome, leading to the proteolytic cleavage of precursors and the release of inflammatory cytokines [195].

**Table 7 ijms-23-02591-t007:** DAMPs in age-related macular degeneration.

Disease	DAMPs	Type	Origin	Localization
AMD 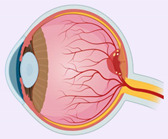	S100A7, S100A8, S100A9 [196]	Ca^2+^ binding protein	Cytoplasmic	Drusen/Retina
	Uric acid [197]	Metabolic product	Cytoplasmic	Serum
	HSP40, HSP60, HSP70, HSP90 & small HSPs [198]	Molecular chaperones	Cytoplasmic	Retina
	ATP [193]	Nucleotide	Cytoplasmic	Vitreous
	Aβ [199]	Peptide	Cytoplasmic	RPE/photoreceptors
	HMGB1 [7] HMGB2 [200]	Nuclear binding protein	Nuclear	RPE Photoreceptor
	IL-1α [201]	Cytokine	Cytoplasmic	RPE
	dsRNA [202]	Nucleic acid	Nuclear	Drusen, RPE
	mtDNA [203]	Nucleic acid	Mitchondria	RPE
	ET-1 [204]	Ribonuclease A	ER	Plasma
	Perlecan [205]	Proteoglycan	Plasma membrane	Retina
	Syndecan-4 [205]	Proteoglycan	Plasma membrane	Retina
	Versican [206]	Proteoglycan	ECM	RPE
	Heparan sulfate [207]	Glycosaminoglycan	ECM	Bruch’s membrane
	Fibronectin [208]	Glycoprotein	ECM	Basal deposits
	Laminin [208]	Glycoprotein	ECM	Basal deposits
	Tenascin-C [209]	Glycoprotein	ECM	CNV membrane

LIF is reported to have a protective role in RPE. It has been characterized as a growth inhibitor and anti-angiogenic molecule, which acts by activating the STAT3 pathway to protect choriocapillaris and possibly prevent atrophy associated with AMD [210]. The presence of S100A7, S100A8, and S100A9 proteins in drusen of AMD patient retinas was confirmed by LC-MS/MS and immunohistochemistry; however, the role of S100 proteins in AMD has not been elucidated [196]. There is a strong relationship between hyperuricemia and AMD, and an increase in serum uric acid was significant in neovascular AMD [197,211]. Increased HSP levels are also observed in the retina of AMD patients, as HSPs regulate protein turnover in the RPE, and thus, provide protection in AMD [198]. However, HSP90 expressed from necrotic RPE cells may function to trigger inflammatory responses in adjacent healthy RPE cells in the retina [212]. HSP70 was proposed as an immunomodulatory protein, as the overexpression of HSP70 significantly suppressed the production of proinflammatory cytokines associated with AMD, along with the elevation of anti-inflammatory cytokines IL-10 and TGF-β1. The extracellular HSP70 exhibits an anti-inflammatory effect by acting on TLR2/TLR4-dependent inhibition of NFκB-driven nuclear translocation [213]. Moreover, intravitreal injection of HSP70 inhibits choroidal neovascularization (CNV)-associated subretinal fibrosis by activation of IL-10 via TLR2/TLR4 receptors [214]. 

With aging, Aβ accumulates at the interface of the RPE and the photoreceptor outer segment in the retina. Subretinal injection of Aβ peptide (1–42) induces retinal inflammation, followed by photoreceptor cell death via endoplasmic reticulum stress [199]. Aβ is one of the key constituents of drusen and causes RPE dysfunction leading to retinal degeneration. It is associated with the activation of microglia, astrocytes, and dendritic cell activation; complement cascade; NFκB pathways; and cytokine production in retinal pigment epithelial cells [215,216]. In the in vitro model of AMD, the RPE cells treated with NaIO_3_, or H_2_O_2_, release HMGB1 from necrotic cells, which can enhance the generation of IL-6 and TNF-α in macrophages and release inflammatory cytokines from RPE cells [7]. Further, HMGB1 activates calveolin-1 and plays an important role in cellular senescence [217]. In the light-induced retinal degeneration animal model, HMGB2 causes photoreceptor cell death by down-regulating nuclear factor erythroid 2-related factor/heme oxygenase-1 (Nrf2/HO-1) and up-regulating NFκB/NLRP3 signaling inflammatory pathways [200]. IL-1α serum concentrations are significantly increased in AMD patients [218,219]. IL-1α released from stressed or dying RPE cells results in the secretion of other pro-inflammatory cytokines. IL-1α is also known to prime the assembly of the NLRP3 inflammasomes in the retina and stimulates the alteration of the cell death profile of damaged RPE cells from apoptosis to pyroptosis, an inflammatory cell death pathway [201]. There was a significant increase in double-stranded (ds)RNA in drusen and RPE in the human eye with geography atrophy [202]. dsRNA enhanced inflammation and neurodegeneration in the retina by receptor-interacting protein (RIP) kinase-dependent necrosis [220]. 

mtDNA damage has been suggested to increase with aging and lesions in RPE cells, mainly from the macular region compared to the periphery. mtDNA damage was positively correlated with the severity of AMD, contrary to the repair capacity. However, the role of released mtDNA from damaged cells in AMD has not been described yet [203]. Although mitochondrial damage was reported in AMD, TFAM changes have not been reported. However, when human monocytic cell lines (THP-1) and human microglia were exposed to rhTFAM, it induced the expression of pro-inflammatory cytokines IL-1β, IL-6, and IL-8 [221]. The adeno-associated virus (AAV)-mediated delivery of calreticulin anti-angiogenic domain (CAD180), along with a functional 112-residue fragment CAD-like peptide 112 (CAD112), to a laser-induced CNV rodent model significantly attenuated neovascularization in mouse eyes. However, the role of calreticulin in AMD needs to be further evaluated [222]. ET-1 significantly increased in the plasma of exudative and neovascular AMD [204]. 

A previous study suggested that the higher expression of HS proteoglycans (HSPGs) in CNV lesions may be linked to endothelial dysfunction and increased capillary permeability [207]. Rat retinas with laser-induced CNV showed significant upregulation of both perlecan and syndecan-4 compared to the control retinas. The expression profiles of these proteoglycans were found not only to depend on the presence or absence of CNV, but also on the size of the CNV-lesion [205]. Intravitreal injection of decorin significantly inhibits laser induced CNV in a rodent model [223]. Decorin might exhibit anti-angiogenic responses by acting as an inhibitor for multiple receptor tyrosine kinases, such as the epidermal growth factor receptor (EGFR), the insulin-like growth factor receptor (IGFR), and the Met hepatocyte growth factor receptor [224]. It has also been shown to decrease hypoxia-induced VEGF expression by blocking the Met expression pathway and downregulating the Ras-related C3 botulinum toxin substrate and HIF1-α [224]. Additionally, the RPE cells with a high-risk genotype at 10q26 for AMD showed significantly enhanced versican expression in the ECM [206]. The localization of HS in Bruch’s membrane (BM) in AMD describes its regulatory role of CNV mainly via its interaction with various angiogenic growth factors, including fibroblast growth factor (FGF), VEGF, TNF-α, TGF-β, and interferon (IFN)-γ [207]. In fact, collagen IV, laminins, and fibronectin are consistently found in the basal deposits of AMD. Since the inhibition of fibronectin matrix assembly in vitro also prevents collagen IV accumulation, it suggests that collagen IV deposition relies on a pre-existing fibronectin matrix [208]. Fibronectin fragments stimulate the release of proinflammatory cytokines, MMPs, and monocyte chemoattractant protein (MCP) from murine RPE cells [225]. There is a strong association between plasma fibrinogen levels and AMD [226]. Tenascin-C is expressed in CNV membranes in eyes with AMD. However, its role in the pathogenesis of CNV remains to be elucidated [227]. Conversely, the intravitreal administration of exogenous sulfated GAGs devoid of core protein was shown to be effective in reducing abnormal retinal or choroidal angiogenesis, implying that the type of core protein bound to GAGs may not be important for their anti-angiogenic effects in vitro [116].

### 2.8. DAMPS in Proliferative Vitreoretinopathy and Rhegmatogenous Retinal Detachment

Proliferative vitreoretinopathy (PVR) is a significant rhegmatogenous retinal detachment (RRD) complication. PVR is characterized by the growth and contraction of cellular membranes within the vitreous cavity resulting in tractional retinal detachment. PVR is primarily driven by fibrotic and inflammatory events involving several DAMPs, which are described in Table 8. 

The DAMPs such as S100, HMGB1, and histones were found to be upregulated in the vitreous of the retinal detachment patients [229,230,234]. Additionally, high levels of S100 protein were observed in the vitreous and epiretinal membranes of PVR and proliferative DR patients, describing its role for the inflammatory axis in the pathogenesis of proliferative retinal disorders [228]. In another study, the PVR subretinal band in patients with chronic recurrent retinal detachment demonstrated pigmented fibrocellular tissue with the foci of cells staining positive for S100 and keratin peripherally, suggesting RPE differentiation [243]. The overexpression of LIF in transgenic mice resulted in pre-retinal membrane formation, contraction, and retinal detachment [130].

HMGB1 plays an essential role in fibrosis in both proliferative DR and PVR. There is a significant increase in HMGB1 in the epiretinal membrane in proliferative DR and PVR [244]. There is also an increase in HMGB1 in the vitreous humor of patients with proliferative DR and retinal detachment compared to patients with retinal detachment alone [245]. Under hypoxia, RPE cells secrete HMGB1. Additionally, HMGB1 can up-regulate the expression of angiogenic and fibrogenic factors in ARPE-19 cells, including VEGF, basic FGF, TGF-β2, and connective tissue growth factor (CTGF), via TLR4 and the RAGE-dependent NFκB pathway [246]. HMGB1 released from the dying cells activates ERK phosphorylation and potentially promotes RPE proliferation and migration, contributing to retinal detachment [230]. HSP47 is linked to increased fibrosis in ARPE-19 cells [231] and significantly inhibits photoreceptor cell death in animal models of retinal detachment [232]. There is significant increased HSP70 expression in the subretinal fibrosis model. HSP90 was found in samples of idiopathic epiretinal membranes, and its expression appears to be correlated with the presence of TGF-β receptor II and αSMA. HSP90 is involved in retinal fibrosis via the TGF-β1-induced transduction pathway in Müller glia [247]. There is a significant increase in extracellular ATP in the vitreous and subretinal space of RRD patients compared to patients with macular holes and epiretinal membranes [233]. In the ATP-induced retinal degeneration feline model, fibrotic tissue ultimately displaced the neural retina in the worst affected area [248]. However, the role of released ATP in fibrosis is not studied yet. Another DAMP member, histone H3, was found on the outer side of the detached retina and was associated with photoreceptor death in the rat model [234]. Additionally, there is a significant increase in IL-1α concentrations in primary RRD subjects due to PVR. Since IL-1 induces RPE cell migration and its intravitreal injection leads to the breakdown of the blood–ocular barrier, IL-1 has been suggested to be an important candidate in the activation processes that lead to PVR development [235]. Müller glia is a primary source of IL-33 in the retina. IL-33 is known for its profibrotic function and increases retinal fibrosis after laser injury [236,249]. IL-33 deficiency enhanced retinal cell death and gliosis after retinal detachment with sustained subretinal inflammation from infiltrating macrophages [250,251]. 

Among proteoglycans, soluble syndecan-1 was significantly high in the vitreous and subretinal fluid collected from RRD eyes. The increase in syndecan-1 concentrations in the subretinal fluid was positively correlated with a longer duration of retinal detachment and negatively correlated with younger age [237]. After retinal detachment, there was a significant increase in biglycan gene expression after seven days of retinal detachment [238]. However, the release of biglycan in the retina or vitreous has not been studied in RD patients or animal models. Further, the hyaluronic acid concentration in the retinal detachment patient was significantly lower than in the control group. Hyaluronidase activity was significantly higher in the vitreous humor of patients with RRD. Contrarily, the vitreous humor contained hyaluronic acid of high molecular mass in the control group [251]. There is also a significant increase in decorin in the epiretinal membrane of PVR and proliferative DR patients [240]. The vitreal decorin concentrations significantly increased in RRD patients who did develop PVR; however, they did not reliably predict the outcome [239]. There is a significant increase in tenascin-C in the epiretinal membrane and vitreous of both proliferative DR and PVR patients [240,241,252]. Tenascin-C is expressed at lower levels in most adult tissues but is transiently upregulated during acute inflammation and is continuously expressed during chronic inflammation and tissue repair [227]. It was predicted to play a role in fibrovascular membrane formation and angiogenesis in proliferative DR [227]. Fibronectin also plays a vital role in retinal detachment. Intravitreal administration of fibronectin and platelet-derived growth factor (PDGF) was sufficient to induce the resultant retinal detachment in the rabbit model [253]. There was a strong correlation between fibrinogen plasma levels and the clinical features of RRD, which supported the role of fibrinogen in retinal detachment [242]. 

### 2.9. DAMPs in Inherited Retinal Disorders

Inherited retinal disorders (IRDs) are a group of rare retinal degenerative disorders that cause severe vision loss due to gene mutations in more than 300 genes, and lead to retinal photoreceptor cell death. IRDs include syndromic forms such as Usher syndrome and non-syndromic forms such as Retinitis Pigmentosa (RP), Leber’s congenital amaurosis, Stargardt’s macular dystrophy, choroideremia, and congenital stationary night blindness [254]. RP is the most common group of IRDs characterized by the slow degeneration of rod and photoreceptors, ultimately leading to the loss of central vision [255]. The DAMPs active during IRDs are described in Table 9.

S100A1 and S100A16 gene expression were significantly high in the Müller glia of retinal-degeneration rd1 mice. S100 proteins are cell-cycle-progression, differentiation, and microtubule-assembly inhibitors, indicating their role in neurodegeneration [256]. In the animal models of retinal degeneration, photoreceptor cell death strongly induces the expression of LIF in a subset of Müller glial cells in the INL of the retina. On the other hand, in the absence of LIF, Müller glial cells remain quiescent and retinal degeneration is enormously accelerated. Further, supplementation of external LIF significantly delays photoreceptor degeneration in the RP model, suggesting their protective role in the retina [67]. Serum uric acid concentrations were significantly high in RP patients and rats with IRD. However, the uric acid content in the retina, brain, and liver was approximately the same as in the controls [257]. Though uric acid has antioxidant properties and plays a neuroprotective role in the brain, its role in RP has yet to be determined.

In IRDs, HSP70 can serve as chaperones against photoreceptor death. The protective role of HSP expression in retinal degenerative disorders has also been confirmed by some laboratory studies, especially concerning oxidative stress [258]. There was a significant increase in the immunoreactivity of Aβ in RGC of eyes with RP, as well as patchy staining of Aβ within sub-RPE deposits, indicating its role in retinal degeneration [259]. In the vitreous humor of RP patients, the HMGB1 level was significantly elevated and associated with necrotic cone-cell death [260]. Additionally, there was a significant increase in HS and chondroitin sulfate in photoreceptor degeneration, irrespective of the IRD model used, similar to their degenerative role in the brain [261,262].

## 3. DAMP-Driven Signal Transduction in Retinal Disorders

### 3.1. RAGE Pathway

DAMPs such as S100 proteins, HMGB1, Aβ, and TFAM act on the RAGE receptor located on the plasma membrane through the adaptor molecule MyD88 (Figure 1) [1]. The interactions of DAMPs and the RAGE signaling pathway have been implicated in an array of retinal disorders such as uveitis, ischemic retinopathies, DR, AMD, and PVR [10,110,228,263]. The interaction of DAMPs with RAGE receptors activates NFκB via AKT, ERK, and p38 signaling pathways, actuating the transcription of cytokines, chemokines and other inflammatory mediators (CCL2, CCL5, CXCL10, CXCL12 TNF-α, IL-1β, IL6, ICAM-1, VCAM-1, NOS-2) [110,263] involved in retinal disorders (Figure 1).

### 3.2. TLR Pathway

The innate immune system is the first line of defense against injury in the retina. DAMPs and PAMPs released by injured retinal cells are recognized by PRRs such as TLRs. The interaction of DAMPS with TLRs has been highly explored in retinal disorders (Figure 2), including endophthalmitis, uveitis, glaucoma, ischemia-reperfusion injury, DR, and AMD [4,11,264,265,266,267]. It is interesting to note that TLR2 function in light-induced retinal degeneration showed sex dependency. In this study, male mice showed significant dependency on TLR2 receptor. The loss of TLR2 in female mice did not impact photoreceptor survival but compromised stress responses, microglial phenotype and photoreceptor survival in male mice [268]. In another study, the treatment of the DNA alkylating agent methyl methanesulfonate induces photoreceptor degeneration in wild-type male mice regulated by poly(ADP-ribose) polymerase 1 (PARP1) activation and cytoplasmic translocation of HMGB1, whereas wild-type female mice are partially protected. Additionally, PARylation was significantly higher in methyl-methanesulfonate-treated male mice and muted in female mice, resulting in enhanced HMGB1 cytoplasmic translocation in male mice. Further, methyl methanesulfonate showed enhanced gliosis and cytokine expression as compared to the retina of female mice [269]. 

### 3.3. NLRP3 Inflammasome

The nucleotide-binding and oligomerization domain NLRs are multi-domain, cytosolic receptors involved in the activation of signaling cascades in ocular disorders. The activation of NLRP3 by DAMPs has been reported in various retinal disorders such as endophthalmitis, uveitis, glaucoma, ischemic retinopathies, DR, AMD, and IRDs [12,74,254,270,271,272,273]. We recently reported the role of NLRP3 inflammasome in proliferative DR [12]. Various DAMPs such as biglycan, LMW hyaluronan, uric acid, and Aβ, after cellular internalization, are processed by the lysozyme. The cathepsin released by this process activates NLRP3 signaling, whereas biglycan, ATP, Aβ, and cathelicidin can directly activate NLRP3 via the P2X7 receptor, a purinergic receptor (Figure 3). DAMPs stimulate inflammasome formations, which are large intracellular multiprotein complexes (MRC) consisting of NLR family sensory proteins (NLRPs), apoptosis speck-like adaptor protein (ASC), and caspase-1 for the production and secretion of IL-1β, leading to further enhancement in photoreceptor cell death by pyroptosis [274]. Additionally, LMW hyaluronan interacts with CD44 and activates the NLRP-3 inflammasome [1,74,198,254,270,271,272,273]. Furthermore, ATP is released by the apoptotic and necrotic cells and acts as a neurotransmitter and as a gliotransmitter in the retina to recruit macrophages and microglia. Once ATP binds P2X7, it activates the protein kinase C/MAP kinase pathway that leads to the release of chemokines and pro-inflammatory cytokines [275]. 

### 3.4. Other Pathways

DAMPs can also activate several other pathways in retinal disorders (Figure 4). The activation of NFκB and AP-1 via CD14 receptors, along with TLR2 or 4, has been reported in various retinal disorders such as endophthalmitis, glaucoma, and DR [1,11,12,70]. Further, the regulation of NFκB and AP-1 by IL-33 and IL-1α via MYD88 has been reported in AMD, glaucoma, DR and PVR [1,201]. Calreticulin and HSP are reported to interact through CD91 and undergo MHC-II antigen representation through proteasomal degradation. Similarly, F-actin interacts through a dendritic-cell-specific receptor (DNGR-1) and undergoes endosomal processing and MHC-1 antigen representation [1]. 

## 4. Therapeutic Implications of DAMPs

In retinal disorders, DAMPs are released by necrotic and apoptotic cells to elicit multiple downstream signaling effects to activate the innate immune system. The emerging evidence from preclinical and clinical studies suggests that DAMPs play both pathogenic and protective roles in retinal disorders. A deeper understanding of the mechanisms of DAMPs will open new opportunities to discover potential biomarkers, therapeutic agents, and therapeutic targets to combat retinal disorders.

### 4.1. DAMPs as Biomarkers

The release of DAMPs may promote chronic and sterile inflammation involved in the pathogenesis of several retinal disorders. Consequently, DAMPs can be valuable diagnostic and prognostic biomarkers in retinal disorders. The potential biomarkers for retinal disorders are tabulated in Table 10. 

DAMPs may have utility in designing treatment modalities [23]. In juvenile idiopathic-arthritis-associated uveitis, measurement of S100 levels from the serum, aqueous humor and tears help to determine the severity/mitigation of the disease. However, the serum may not provide representative local inflammation, and access to the aqueous humor may not be viable. On the contrary, tears are easily accessible and the development of assays/methods to measure S100 may offer a more precise way to quantify disease activity, in addition to the current grading of anterior chamber cells by Standardization of Uveitis Nomenclature criteria [30]. The diagnosis of Behcet’s disease (BD) uveitis in early stages has been problematic, which may be resolved by measuring serum levels of HSP-70 for BD uveitis [32]. Similarly, the measurement of serum S100A8/S100A9 concentrations in the various stages of diabetic retinopathy could provide a greater clue for the progression of the disease [5]. DAMP biomarkers may provide earlier diagnoses and risk assessments, possibly catering to safe, personalized treatment to individual patients. However, the major limitations to their application as biomarkers could be their versatile nature and activation in multiple diseases. For example, HMGB1 levels were increased in the vitreous of endophthalmitis, IR, DR, PVR/RRD, and IRDs (Table 10). Therefore, a secondary diagnostic procedure or signature panel of DAMPs might be instrumental to identify retinal disorders.

### 4.2. DAMPs as Therapeutic Targets

The excessive production of DAMPs in response to infection, inflammation, or injury has led to the discovery of several proteins and molecules that can be targeted to develop novel therapeutics or repurpose existing drugs to treat retinal disorders. The potential drug targeting DAMPs suggested in the retinal disorders are tabulated in Table 11. Exploring DAMPs as therapeutic targets for developing new treatments for chronic disorders, including DR, AMD, glaucoma and PVR, involves tight regulation of the immune responses to retinal injury. In addition, DAMP-targeted therapies could enable the modulation of excessive inflammatory cascade triggered during the sterile inflammation. For example, glycyrrhizin reduces diabetes-induced neuronal and vascular damage by inhibiting inflammation, specifically by activating HMGB1 through the sirtuin 1 (SIRT1) pathway. Similarly, uveitis may be targeted by inhibiting interphotoreceptor retinoid-binding-protein (IRBP)-specific T cell proliferation and their IFN-γ and IL-17 production [276]. Further, DAMPs such as HMGB1 have been studied as a therapeutic target in multiple retinal disorders, including antibody-based therapies; protein, oligonucleotide, and small molecule inhibitors; blockage of HMGB1-receptor signaling; and targeting with miRNAs [31,276,277,278]. However, special consideration is warranted for considering DAMPs as therapeutic targets. It is essential to discriminate the deleterious role of the DAMPs (usually long-term), contrary to the innate immune responses initiated in the early stage of the disease progression (protective role). Recently, we found increased aqueous humor decorin concentrations associated with the progression of diabetic retinopathy [157]. Additionally, the most effective way to target DAMPs in the retina is cell- and location-dependent (intracellular or extracellular). Thus, long-term targeting of DAMPs may prevent them from initiating regulatory T cells (Tregs) and promote immunosuppression [279].

### 4.3. DAMPs as Therapeutic Agents

DAMPs such as decorin, LIF, defensins, IL-33, syndecan-1, and SAA have been described for their anti-inflammatory, anti-angiogenic or anti-fibrotic properties, and are presented in Table 12. Harnessing these DAMPs as therapeutic agents or therapeutics is an attractive novel therapeutic strategy against retinal disorders, which will perhaps find its way into future routine treatment modalities. The availability of DAMPs during injury or infection is vital for providing the essential host defense system and restoring homeostasis in the injured tissues. Cathelicidins act as an anti-microbial agent on many pathogens, including Gram-positive and Gram-negative bacteria, fungi, parasites, and enveloped viruses in vitro. Cationic cathelicidins can bind and disrupt negatively charged membranes, leading to microbial cell death. These peptides can also cross membranes and target intracellular processes such as RNA and DNA synthesis, impair the functions of enzymes and chaperones, and can stimulate protein degradation [285]. However, under physiological circumstances, most cathelicidins are impaired by high salt concentrations, sugars, and other host or microbial factors [285]. IL-33 is upregulated in the uveitis retina depicting its anti-inflammatory role. However, its effects are model- and disease-specific. Thus, considering IL-33 as a therapeutic agent needs a complete understanding of its function in the pathological microenvironment [45]. HSP70 binds to TLR2/TLR4 and exhibits anti-inflammatory properties via secretion of IL-10 and TGF-β in AMD. Contrarily, the extracellular HSP70 has been suggested to have a pro-inflammatory effect. Hence, it is too early to predict the use of HSP-70 as an anti-inflammatory agent. Decorin binds to TGFβ with high affinity and is known for its anti-fibrotic role compared to traditional anti-fibrotic adjuvants such as mitomycin-C and 5-fluorouracil [286,287]. Nevertheless, it has both pro-and anti-angiogenic properties [224]. Similarly, anti-angiogenic therapeutic DAMPs such as LIF, HS, and IL-33 are also in the early stages and their safety and efficacy profile is awaited for retinal disorders. Therefore, the application or inhibition of DAMPs must be designed under strict caveats and precautions using the therapeutic window during the progression of retinal disorders. 

## 5. Conclusions and Future Directions

In this comprehensive review, we have: (i) described the role of DAMPs in various retinal disorders; (ii) demonstrated that DAMP-driven signaling pathways are involved in the pathogenesis of retinal disorders, and iii) discussed the possibility of DAMPs acting as biomarkers, therapeutic targets, and therapeutic agents for the management of vision-threatening retinal disorders. 

Epigenetic mechanisms have emerged as critical modulators of the host defense system in the retina. Epigenetic modifications have been implicated in various retinal disorders, including uveitis, glaucoma, ocular cancer, IR, DR, AMD, PVR, RRD, and IRDs [290]. Though the role of DAMPs in epigenetic modifications in retinal diseases has not been directly evaluated, excessive or persistent DAMP-mediated signaling cascades may initiate epigenetic changes in chronic retinal diseases such as DR, AMD, and PVR [143,213,250,291]. Additionally, following severe or prolonged damage, a loss of intracellular DAMPs increases genomic instability and may cause epigenetic alteration [292]. DAMPs such as biglycan, ATP, Aβ, and cathelicidin can directly activate NLRP3 via the P2X7 receptor, promoting the inflammatory response (Figure 3). The expression of the P2X7 receptor is controlled via promoter methylation in neurodegenerative diseases [293]. The extracellular HMGB1 interacts with RAGE and TLR receptors in retinal diseases (Figure 1 and Figure 2) to actuate inflammatory pathways [228,264]. HMGB1 may act as an epigenetic modifier that leads to the silencing of TNF-α and IL-1β responses [294]. Therefore, future in-depth studies are required to completely understand the epigenetic changes caused by DAMPs in retinal disorders. The diverse nature of the retinal cell types and their neuronal circuity complicates our understanding of the cell-specific immune responses and the release of DAMPs in various retinal disorders. Therefore, future studies are warranted to identify the DAMPs involved in the molecular mechanisms of retinal diseases, employing single-cell or cell-specific proteomic signatures to identify/design or repurpose next generation therapeutics for retinal disorders.

## Figures and Tables

**Figure 1 ijms-23-02591-f001:**
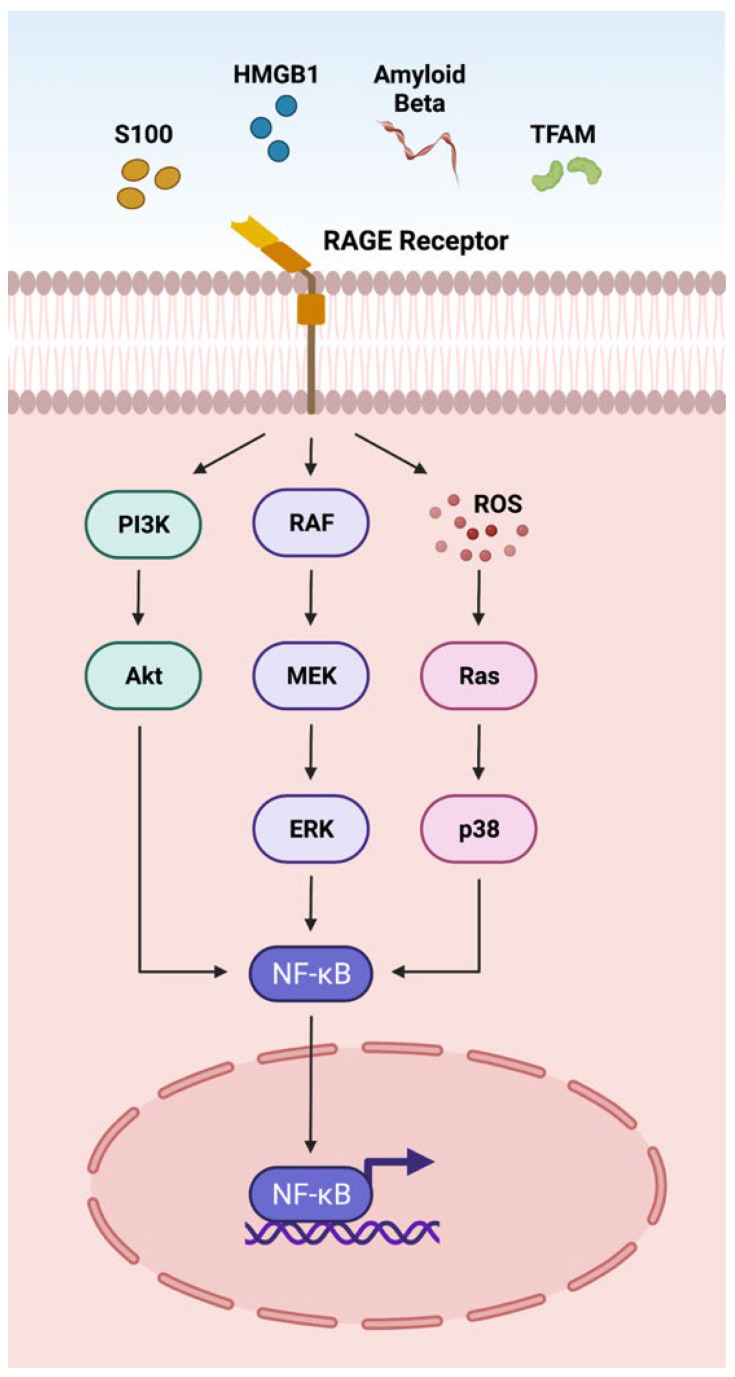
Overview of the DAMPs activating the RAGE pathway. The receptor for advanced glycation end-products (RAGE) is a multi-ligand protein that integrates the immunoglobulin superfamily of receptors. RAGE recognizes a variety of DAMPs including S100, high mobility group box 1 protein (HMGB1), Amyloid beta (Aβ), and transcription factor A mitochondrial (TFAM). RAGE activation leads to downstream NFκB signaling and transcription of inflammatory factors.

**Figure 2 ijms-23-02591-f002:**
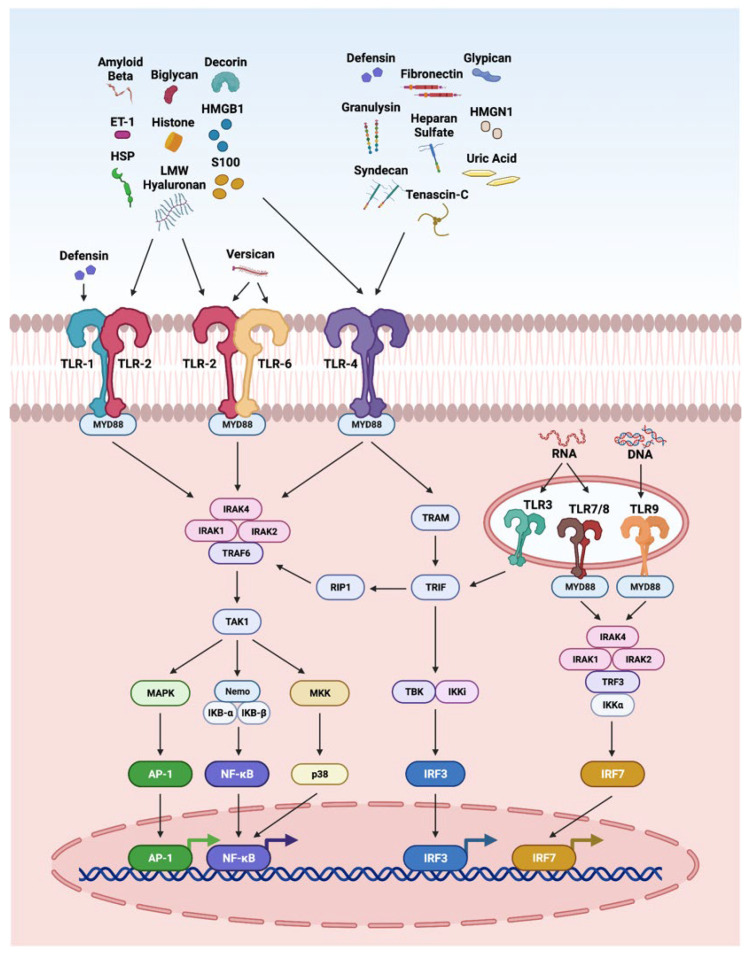
Overview of DAMPs activating the toll-like receptor (TLR) pathways. TLRs recognize a variety of DAMPs. Defensin activates TLR1; biglycan, decorin, versican, LMW hyaluronan, S100, HSP, Aβ, histones, HMGB1, and ET-1 activate TLR2; biglycan, decorin, LMW hyaluronan, HS, fibronectin, tenascin-C, S100, HSP, uric acid, Aβ, histones, HMGB1, HMGN1, ET-1, defensins, granulysin, syndecan, and glypican are reported to activate TLR4; versican activates TLR6; RNA activates TLR3, 7 and 8; and DNA activates TLR9. When TLRs are stimulated by DAMPs they dimerize and recruit downstream adaptor molecules, such as myeloid differentiation primary-response protein 88 (MyD88), and TRIF-related adaptor molecule (TRAM), which directs downstream molecules, leading to the activation of signaling cascades that converge at the NFκB, activator protein 1 (AP1), and interferon response factors (IRFs).

**Figure 3 ijms-23-02591-f003:**
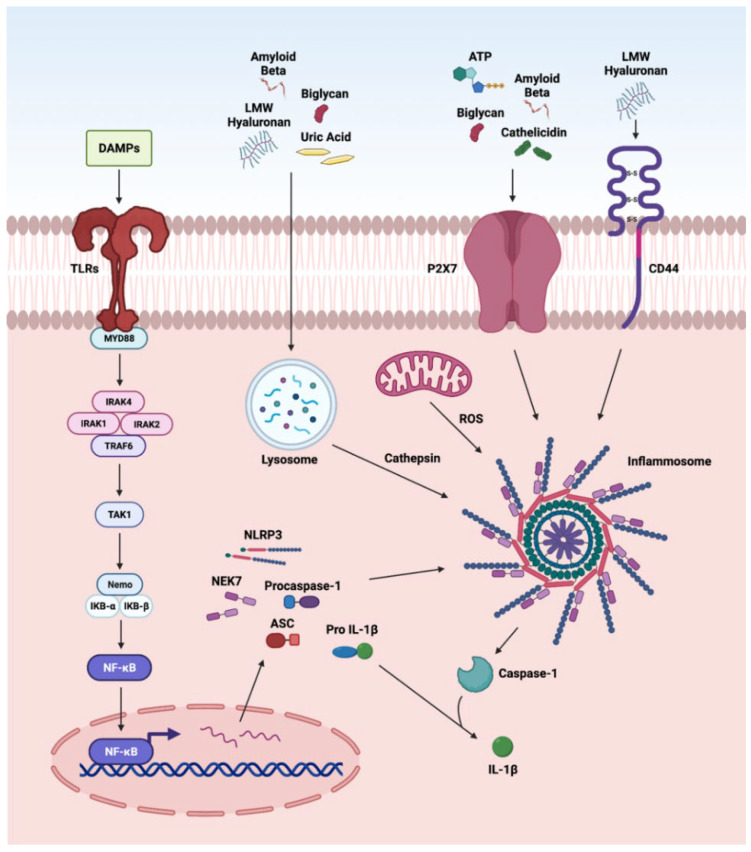
Overview of DAMPs activating the NLRP3 inflammasome. The first step in the 2-step process involves activation and translocation of NFκB into the nucleus to regulate the transcription of the oligomerization-like receptor pyrin-domain-containing protein 3 (NLRP3) inflammasome genes. The second step is actuation of NLRP3 inflammasome mediated by (a) DAMPs such as biglycan, LMW hyaluronan, uric acid, and Aβ to release cathepsin from lysosomal degradation; (b) K+ efflux via P2X7 receptor activation by DAMPs such as biglycan, ATP, Aβ, and cathelicidin; and (c) CD44 activation by LMW hyaluronan caspase-1 signaling pathway leading to caspase-1 activity and the release of mature IL-1β.

**Figure 4 ijms-23-02591-f004:**
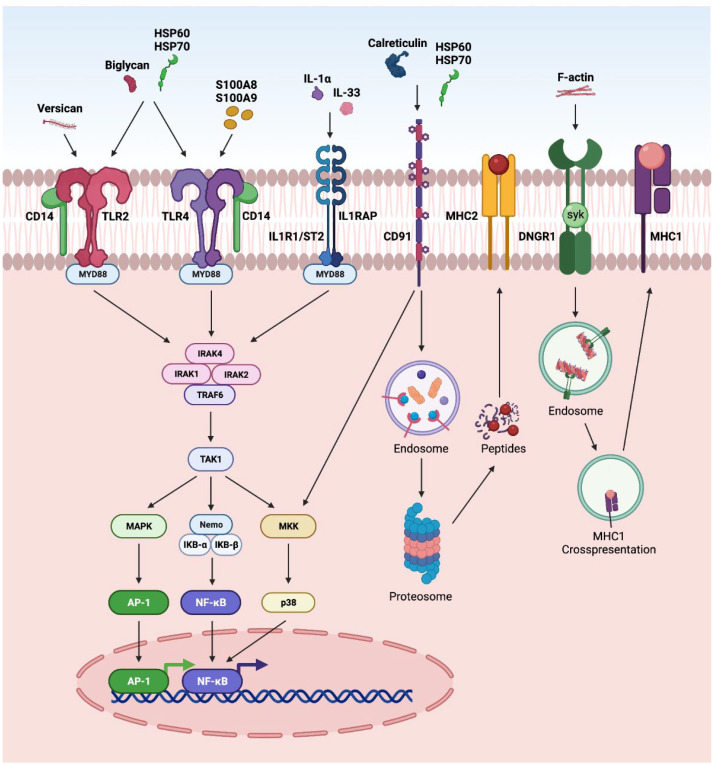
Overview of DAMPS activating other pathways: (a) In CD14-dependent pathway, DAMPs such as biglycan, versican, HSP60, and HSP70 activate TLR2, whereas DAMPs such as biglycan, HSP60, HSP70, S100A8, and S100A9) activate TLR4 signaling. After TLRs become activated, they dimerize and recruit downstream adaptor molecules, such as myeloid differentiation primary-response protein 88 (MyD88), initiating downstream signaling cascades that converge at NFκB and AP1 and leading to the transcription of inflammatory factors; (b) in ILR1/ST2 signaling pathway, DAMPs such as IL-1α and IL-33 can signal through IL1R1/IL1RAP. IL-1 or IL-33 activate the heterodimeric signaling receptor complex formation of IL1R1/IL1RAP, which creates the scaffold for MyD88 dimerization converging to NFκB pathway; (c) in CD91 signaling pathway, DAMPs such as calreticulin, HSP60, and HSP70 interact with CD91, which leads to endocytosis of calreticulin or HSPs and proteosome degradation, and cross-presentation of the chaperoned antigens culminating in co-stimulation of T cells; (d) in DNGR1 signaling pathway, F-actin interacts with DNGR1, which signals through the spleen tyrosine kinase (SYK), diverting phagocytosed cargo toward endosomal compartments, leading to cross-presentation and generation of resident memory CD8+ T cells.

**Table 1 ijms-23-02591-t001:** DAMPs in endophthalmitis.

Disease	DAMPs	Type	Origin	Localization
Endophthalmitis 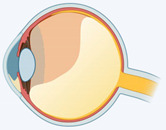	S100A7, S100A9 [18]	Ca^2+^ binding protein	Cytoplasmic	Retina
	HMGB1 [20]	Nuclear binding protein	Nuclear	Vitreous
	αβ-crystallin [21]	Molecular chaperones	Cytoplasmic	Retina
	LIF [22]	Cytokines	Cytoplasmic	Retina
	IL-1α [23]	Cytokines	Cytoplasmic	Vitreous
	β−defensin-1, -2 [24,25]	Antimicrobial protein	ER	RPE/CBE/Müller glia
	Cathelicidin LL37 [26]	Antimicrobial protein	ER	Müller glia
	SAA [27]	Acute-phase protein	Plasma	Serum

**Table 2 ijms-23-02591-t002:** DAMPs in uveitis.

Disease	DAMPs	Type	Origin	Localization
Uveitis 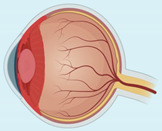	S100A8, S100A9, S100A12 [30]	Ca^2+^ binding protein	Cytoplasmic	Serum/aqueous/tears
	HMGB1 [31]	Nuclear binding protein	Nuclear	Retina
	HSP70 [32]	Molecular chaperones	Cytoplasmic	Serum
	SAA [33]	Acute-phase protein	Plasma	Aqueous
	Fibronectin [34]	Glycoprotein	ECM	Iris
	Fibrinogen [35]	Glycoprotein	ECM	Iris

**Table 3 ijms-23-02591-t003:** DAMPs in glaucoma.

Disease	DAMPs	Type	Origin	Localization
Glaucoma 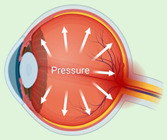	S100B [46]	Ca^2+^ binding protein	Cytoplasmic	Astrocyte/Müller glia
	LIF [47]	Cytokine	Cytoplasmic	Retina
	Uric acid [48]	Metabolic product	Cytoplasmic	Serum
	HSP60, HSP70 [49]	Molecular chaperones	Cytoplasmic	Retina
	ATP [50]	Nucleotide	Cytoplasmic	Aqueous/vitreous
	Aβ [51]	Peptide	Cytoplasmic	Aqueous/optic nerve
	Histone-H4 [52]	Nuclear binding protein	Nuclear	Serum
	HMGB1 [53]	Nuclear binding protein	Nuclear	Aqueous
	IL-1α [54]	Cytokine	Cytoplasmic	Aqueous
	mtDNA [55]	Nucleic acid	Mitchondria	Ganglion cell
	Calreticulin [56]	Multifunction soluble protein	ER	Nerve fiber layer
	ET-1 [57]	Ribonuclease A	ER	Astrocyte
	Decorin [58]	Proteoglycan	ECM	Aqueous
	Biglycan [59]	Proteoglycan	ECM	Optic nerve
	Versican [60]	Proteoglycan	ECM	Trabecular meshwork
	Aggrecan [61]	Proteoglycan	ECM	Optic nerve
	Phosphocan [62]	Proteoglycan	ECM	Retina/optic nerve
	HS [63]	Glycosaminoglycan	ECM	Retina/trabecular meshwork
	Fibronectin [62,64]	Glycoprotein	ECM	Retina/optic nerve
	Laminin [62]	Glycoprotein	ECM	Retina/optic nerve/astrocytes
	Tenascin-C [62]	Glycoprotein	ECM	Trabecular meshwork
	SAA [65]	Acute-phase protein	Plasma	Trabecular meshwork and plasma

**Table 4 ijms-23-02591-t004:** DAMPs in ocular cancer.

Disease	DAMPs	Type	Origin	Localization
Ocular cancer 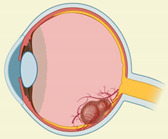	S100 [83]	Ca^2+^ binding protein	Cytoplasmic	Astrocytes/ganglion cell/Müller glia
	S100A1 [86]	Ca^2+^ binding protein	Cytoplasmic	Uveal melanoma
	S100B [87]	Ca^2+^ binding protein	Cytoplasmic	Serum
	Uric acid [88]	Metabolic product	Cytoplasmic	Aqueous
	HSP70, HSP90 [89,90]	Molecular chaperones	Cytoplasmic	Retina/extracellular vesicles
	HMGB1 [91]	Nuclear binding protein	Nuclear	Retinoblastoma
	cfcDNA [92]	Nucleic acid	Nuclear	Plasma
	Versican [93]	Proteoglycan	ECM	Uveal melanoma

**Table 5 ijms-23-02591-t005:** DAMPs in ischemic retinopathies.

Disease	DAMPs	Type	Origin	Localization
Ischemic retinopathy 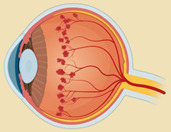	S100 [103], S100A4 [104]	Ca^2+^ binding proteins	Cytoplasmic	Ganglion cell
	Uric acid [105]	Metabolic product	Cytoplasmic	Retina
	HSP27, HSP60, HSP70, αβ-crystallin [106,107,108]	Molecular chaperones	Cytoplasmic	Retina (RGC, RPE, INL)
	Cyclophilin A [109]	Ubiquitous protein	Cytoplasmic	Neuron
	LIF [47]	Peptide	Cytoplasmic	Retina
	HMGB1 [110]	Nuclear binding protein	Nuclear	Vitreous/retina
	IL-1α [111,112]	Cytokine	Cytoplasmic	Retina/plasma
	TFAM [113,114]	Transcription factor	Mitchondria	Retina (OPL, INL, IPL, GCL)
	Decorin [115]	Proteoglycan	ECM	Retina (INL)
	Fibronectin [115]	Glycoprotein	ECM	Retina
	Laminin [115]	Glycoprotein	ECM	Optic nerve
	Tenascin-C [115]	Glycoprotein	ECM	Optic nerve
	HS [116]	Glycosaminoglycan	ECM	Optic nerve
	Chondritin sulfate [115]	Glycosaminoglycan	ECM	Optic nerve
	Aggrecan [115]	Proteoglycan	ECM	Optic nerve

**Table 8 ijms-23-02591-t008:** DAMPs in proliferative vitreoretinopathy and rhegmatogenous retinal detachment.

Disease	DAMPs	Type	Origin	Localization
PVR/RRD 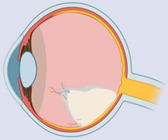	LIF [130]	Peptide	Cytoplasmic	Pre-retinal membrane
	S100 [228,229]	Ca^2+^ binding protein	Cytoplasmic	Epiretinal membrane/subretinal fluid
	HMGB1 [230]	Nuclear binding protein	Nuclear	Vitreous
	HSP47, HSP70 [231,232]	Molecular chaperones	Cytoplasmic	RPE/inner segments
	ATP [233]	Nucleotide	Cytoplasmic	Vitreous/subretinal fluid
	Histone-H3 [234]	Nuclear binding protein	Nuclear	Vitreous/ detached retina
	IL-1α [235]	Cytokine	Cytoplasmic	Subretinal fluid
	IL-33 [236]	Cytokine	Cytoplasmic	Müller glia
	Syndecan-1 [237]	Proteoglycan	Plasma membrane	Vitreous/subretinal fluid
	Biglycan [238]	Proteoglycan	ECM	Retina
	Decorin [239,240]	Proteoglycan	ECM	Vitreous/epiretinal membrane
	Tenascin-C [240,241]	Glycoprotein	ECM	Vitreous/epiretinal membrane
	Fibrinogen [242]	Glycoprotein	ECM	Plasma

**Table 9 ijms-23-02591-t009:** DAMPs in inherited retinal diseases.

Disease	DAMPs	Type	Origin	Localization
IRD 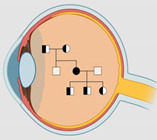	S100A1, S100A16 [256]	Ca^2+^ binding protein	Cytoplasmic	Müller glia
	LIF [67]	Peptide	Cytoplasmic	Müller glia
	Uric acid [257]	Metabolic product	Cytoplasmic	Serum
	HSP70 [258]	Molecular chaperones	Cytoplasmic	Photoreceptors
	Aβ [259]	Peptide	Cytoplasmic	GCL/sub-RPE deposits
	HMGB1 [260]	Nuclear binding protein	Nuclear	Vitreous
	HS [261,262]	Glycosaminoglycan	ECM	Photoreceptors
	Chondritin sulfate [261,262]	Glycosaminoglycan	ECM	Photoreceptors

**Table 10 ijms-23-02591-t010:** DAMPs as biomarkers.

Disorders	DAMPs	Plasma/ Serum	Tear	Vitreous	Aqueous	Refs
Endophthalmitis	HMGB1	✕	✕	✓	✕	[20]
	IL-1α	✕	✕	✓	✕	[23]
	SAA	✓	✕	✕	✕	[27]
Uveitis	S100	✕	✓	✕	✕	[30]
	HSP	✓	✕	✕	✕	[32]
	SAA	✕	✕	✕	✓	[33]
Glaucoma	Uric acid	✓	✕	✕	✕	[48]
	ATP	✕	✕	✓	✓	[50]
	Aβ	✕	✕	✕	✓	[51]
	Histone	✓	✕	✕	✕	[52]
	HMGB1	✕	✕	✕	✓	[53]
	IL-1α	✕	✕	✕	✓	[54]
	Decorin	✕	✕	✕	✓	[58]
	SAA	✓	✕	✕	✕	[65]
Ocular cancer	Uric acid	✕	✕	✕	✓	[88]
	DNA	✓	✕	✕	✕	[92]
Ischemic Retinopathies	HMGB1	✕	✕	✓	✕	[110]
	IL-1α	✓	✕	✕	✕	[112]
Diabetic Retinopathy	S100A8, S100A9	✓	✕	✕	✕	[5]
	HMGB1	✕	✕	✓	✕	[148]
	Uric acid	✓	✕	✓	✕	[149]
	Cyclophilin A	✓	✕	✕	✕	[151]
	Cathelicidin	✓	✕	✕	✕	[154]
	Defensins	✓	✕	✕	✕	[155]
	Syndecan	✓	✕	✕	✕	[156]
	Decorin	✓	✕	✕	✓	[157,158]
	Versican	✓	✕	✕	✕	[159]
	LMW hyaluronan	✕	✓	✓	✕	[160]
	HS	✕	✕	✓	✕	[160]
	Fibronectin	✓	✕	✓	✓	[34]
	Fibrinogen	✓	✕	✕	✕	[164]
	Tenascin-C	✕	✕	✓	✕	[165]
AMD	Uric acid	✓	✕	✕	✕	[197]
	ATP	✕	✕	✓	✕	[193]
	ET-1	✓	✕	✕	✕	[204]
PVR/RRD	HMGB1	✕	✕	✓	✕	[19]
	ATP	✕	✕	✓	✕	[233]
	Histone	✕	✕	✓	✕	[229,234]
	Syndecan	✕	✕	✓	✕	[237]
	Decorin	✕	✕	✓	✕	[239]
	Tenascin-C	✕	✕	✓	✕	[241]
	Fibrinogen	✓	✕	✕	✕	[242]
IRD	Uric acid	✓	✕	✕	✕	[257]
	HMGB1	✕	✕	✓	✕	[260]

The DAMPs are represented by ✓ for their presence and ✕ for their absence.

**Table 11 ijms-23-02591-t011:** DAMPs as therapeutic targets.

Retinal Disorders	Drug	DAMPs	Action	Refs
Uveitis	HMGB1 Ab Glycyrrhizin	HMGB1	Inhibition of IRBP-specific T cell proliferation	[31]
Glaucoma	Brilliant Blue G and N-methyl-d-aspartic acid	ATP	Antagonists of the P2X7R	[280]
Ocular cancer	Ansamycins	HSP90	G1 Arrest	[281]
	miR34A and miR-22	HMGB1	Act on autophagy, migration, and invasion of RB cells	[277,278]
Ischemic retinopathies	Cyclosporin A	TFAM	Preserves TFAM	[113]
	Coenzyme Q10	TFAM	Preserves TFAM	[282]
	Brimonidine	TFAM	Preserves TFAM	[114]
Diabetic retinopathy	Tasquinimod Glycyrrhizin	S100 HMGB1	Inhibits angiogenesis via TSP-1, VEGF, ICAM-1 and ERK1/2 Acts on HMGB1 via SIRT1 and provides neurovascular protection	[276,283]
AMD	Geldanamycin	HSP90	Inhibits VEGF and HIFα	[284]
PVR/RRD	Anti-histone Antibodies	Histone	Reduced retinal damage	[234]
	Geranylgeranylacetone	HSP70	Activation of Akt pathway	[232]

**Table 12 ijms-23-02591-t012:** DAMPs as therapeutic agents.

Retinal Disorders	DAMPs	Therapeutic Property	Signaling Pathway	Refs
Endophthalmitis	Cathelicidin	Anti-microbial	TLRs	[26]
Uveitis	IL-33	Anti-inflammatory	Macrophage M2 polarization	[45]
Glaucoma	Decorin	Anti-fibrotic	Inhibits TGF-β	[286]
Diabetic retinopathy	LIF HS	Anti-angiogenic Anti-angiogenic	HIF-1α and VEGF Inhibiting VEGF-VEGFR2 binding	[116,288]
AMD	LIF	Anti-angiogenic	STAT3 pathway	[210]
	HSP70	Anti-inflammatory	TLR2/TLR4	[213,214]
	IL-33	Anti-angiogenic	Not known	[289]

## Data Availability

Data are available on request.

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
