# Peer review of "Damage-Associated Molecular Patterns (DAMPs) in Retinal Disorders"

_ijms, 2022, doi:10.3390/ijms23052591_

Round 1

Reviewer 1 Report

I found the manuscript "Damage-Associated Molecular Patterns (DAMPs) in Retinal Disorders" well built. The impressive literature investigation is clearly visible, and also I really appreciated the tables and figures which really helped focusing on the content. I have a minor and a major comment, as follows:

1) authors should check in the template for the journal where citations should be added respect to punctuation. I think they should be in the format [n]., please check and correct.

2) I feel that chapter 4 was not properly described. The topic is important and it can really help other scientists correctly address this aspect in their research. I appreciate the tables, but authors should expand the commentary on limitations and possibilities in the use of DAMPs as biomarkers, therapeutic agents or targets. Addressing this second point is crucial to increase the impact of the manuscript.

Author Response

Reviewer-1

I found the manuscript "Damage-Associated Molecular Patterns (DAMPs) in Retinal Disorders" well built. The impressive literature investigation is clearly visible, and also, I really appreciated the tables and figures which really helped focusing on the content.

Thank you.

Comment 1: authors should check in the template for the journal where citations should be added respect to punctuation. I think they should be in the format [n]., please check and correct.

We apologize for the mistake. The text is formatted using the IJMS template.

Comment 2: I feel that chapter 4 was not properly described. The topic is important, and it can really help other scientists correctly address this aspect in their research. I appreciate the tables, but authors should expand the commentary on limitations and possibilities in the use of DAMPs as biomarkers, therapeutic agents or targets. Addressing this second point is crucial to increase the impact of the manuscript.

The authors appreciate the reviewer’s suggestions. As per the suggestions, we have expanded chapter 4 and included the possibilities and limitations of DAMPs as biomarkers, therapeutic agents, and targets. This has greatly enhanced the manuscript.

Reviewer 2 Report

This is a well-written and organized paper that flows smoothly and is appealing to read. It covers an important topic. The breadth and depth are appropriate.

This reviewer is in support of accepting this manuscript after the following minor comments are addressed:

  1. Sex-dependence is not discussed, particularly because there is evidence to suggest that activation of TLR2 protects both male and female mice from light damage, while downregulation of TLR2 in female mice does not impact photoreceptor survival. Is there any interplay of DAMPs in a sex-dependent fashion?
  2. While discussing HMGB and DAMPs, the following reference was missed: Free Radic Biol Med. 2022 Jan 25;181:14-28. doi: 10.1016/j.freeradbiomed.2022.01.018. Online ahead of print.
  3. The interplay between epigenetics, DAMPs and retinal disease is overlooked. While this reviewer is aware that the literature is scarce on this area, it is notheless an important aspect to be discussed and perhaps addressed as a future perspective.
  4. Can proteomic approaches be utilized to help advance this area of DAMPs in the context of retinal diseases? Would the authors hypothesize on this in a section that they can add and entitle as “future perspectives”?
  5. The cytoplasmic double stranded DNA in Fig 2 does not appear to circular like mtDNA, and its source/function is not well-elaborated on in the text.

Very Minor:

Some typos.

Reference number is sometimes added after the “period/fullstop” and sometimes before. Please use the journal’s guidelines.

Figures legends need to be much more extensive than a simple title.

NFKB in figure 1 cannot be in the nuclear membrane. It shall be as drawn in Fig 2.

Author Response

Reviewer-2

This is a well-written and organized paper that flows smoothly and is appealing to read. It covers an important topic. The breadth and depth are appropriate. This reviewer is in support of accepting this manuscript after the following minor comments are addressed.

Thank you.

Comment 1: Sex-dependence is not discussed, particularly because there is evidence to suggest that activation of TLR2 protects both male and female mice from light damage, while downregulation of TLR2 in female mice does not impact photoreceptor survival. Is there any interplay of DAMPs in a sex-dependent fashion?

The authors thank the reviewer’s suggestions. Though the literature is meager, the sex dependency has been added in section 3.2

Comment 2: While discussing HMGB and DAMPs, the following reference was missed: Free Radic Biol Med. 2022 Jan 25;181:14-28. doi: 10.1016/j.freeradbiomed.2022.01.018. Online ahead of print.

We have included the suggested reference in section 2.7

Comment 3: The interplay between epigenetics, DAMPs and retinal disease is overlooked. While this reviewer is aware that the literature is scarce on this area, it is notheless an important aspect to be discussed and perhaps addressed as a future perspective.

We agree with the reviewer. We have included the possible interplay between epigenetics, DAMPs and retinal disorders in section 5 (Conclusions and Future Directions)

Comment 4: Can proteomic approaches be utilized to help advance this area of DAMPs in the context of retinal diseases? Would the authors hypothesize on this in a section that they can add and entitle as “future perspectives”?

Added in section 5

Comment 4: The cytoplasmic double-stranded DNA in Fig 2 does not appear to circular like mtDNA, and its source/function is not well-elaborated on in the text.

Fig 2 is corrected and source/function is mentioned in the text as suggested

Comment 5: Reference number is sometimes added after the “period/fullstop” and sometimes before. Please use the journal’s guidelines.

We apologize for the mistake. The correction is incorporated in the revised manuscript.

Comment 6: Figures legends need to be much more extensive than a simple title.

The expanded figure legends are incorporated in the revised manuscript.

Comment 7: NFKB in figure 1 cannot be in the nuclear membrane. It shall be as drawn in Fig 2.

We agree with the reviewer. We have corrected figure 1 accordingly.

Round 2

Reviewer 1 Report

I really appreciate the manuscript in its revised form. I recommend publication.